# AN INFORMATION CRITERION FOR CONTROLLED DISENTANGLEMENT OF MULTIMODAL DATA

**Chenyu Wang**[*1,2], **Sharut Gupta**[*1], **Xinyi Zhang**[1,2], **Sana Tonekaboni**[2],

**Stefanie Jegelka**[1,3], **Tommi Jaakkola**[1], **Caroline Uhler**[1,2]

[1]MIT  [2]Broad Institute of MIT and Harvard  [3]TU Munich

## ABSTRACT

Multimodal representation learning seeks to relate and decompose information inherent in multiple modalities. By disentangling modality-specific information from information that is shared across modalities, we can improve interpretability and robustness and enable downstream tasks such as the generation of counterfactual outcomes. Separating the two types of information is challenging since they are often deeply entangled in many real-world applications. We propose **Disentangled Self-Supervised Learning** (DISENTANGLEDSSL), a novel self-supervised approach for learning disentangled representations. We present a comprehensive analysis of the optimality of each disentangled representation, particularly focusing on the scenario not covered in prior work where the so-called *Minimum Necessary Information* (MNI) point is not attainable. We demonstrate that DISENTANGLEDSSL successfully learns shared and modality-specific features on multiple synthetic and real-world datasets and consistently outperforms baselines on various downstream tasks, including prediction tasks for vision-language data, as well as molecule-phenotype retrieval tasks for biological data. The code is available at https://github.com/uhlerlab/DisentangledSSL.

## 1 INTRODUCTION

Humans understand and interact with the world using multiple senses, each providing unique and complementary information essential for forming a comprehensive mental representation of the environment. To emulate human-like perception, multimodal representation learning (Bengio et al., 2013; Liang et al., 2022a) seeks to decipher complex systems by combining information from multiple modalities into holistic representations, showcasing significant applications across fields from vision-language (Yuan et al., 2021; Lu et al., 2019; Radford et al., 2021) to biology (Yang et al., 2021; Zhang et al., 2022; Wang et al., 2023a). Large multimodal representation learning models such as CLIP (Radford et al., 2021), trained through self-supervised learning, maximally capture the mutual information shared across multiple modalities. These models exploit the assumption of multi-view redundancy (Tosh et al., 2021; Sridharan & Kakade, 2008), which indicates that shared information between modalities is exactly what is relevant for downstream tasks.

However, the misalignment between modalities restricts the application of these methods in real-world multimodal scenarios. Among various contributing factors, the modality gap, rooted in the inherent differences in representational nature and information content across modalities (Liang et al., 2022b; Ramasinghe et al., 2024; Huh et al., 2024), plays a significant role. This highlights the need for a *disentangled representation space* that captures both shared and modality-specific information:

- **Coverage:** Distinct modalities often contribute unique, complementary information crucial for specific tasks (Liang et al., 2024; Liu et al., 2024b), making it essential to capture modality-specific features effectively. For instance, in multimodal sentiment analysis, text conveys explicit sentiment, while vocal tone and facial expressions provide nuanced emotional cues.

- **Disentanglement:** As emphasized by Zhang et al. (2024) in biological contexts, separating shared from modality-specific information is vital for interpretability and decision-making. However,

---

[*]Equal contribution

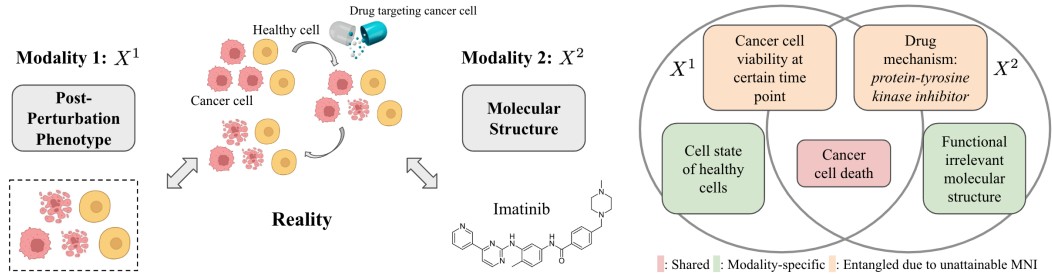

Figure 1: Post-perturbation phenotype ($X_1$) (i.e., cellular images or gene expression after the application of a drug to cells) and molecular structure ($X_2$) of an underlying drug perturbation system where cancer cells are targeted and killed while healthy cells remain unaffected. The Venn diagram illustrates shared and specific information between modalities $X_1$ and $X_2$: shared content is shown in red, modality-specific content in green, and entangled content due to unattainable MNI in orange. For example, for the drug mechanism, the molecular structure conveys full information, while the phenotype provides partial information (i.e. mechanisms causing cell death). Similarly, for the states of healthy cells, the phenotype specifies their cell states, whereas the molecular structure only indicates that the cells are unaffected without detailing their specific states.

maximizing mutual information across modalities can blur this separation (Fischer, 2020), reducing stability and robustness for tasks like cross-modal translation and counterfactual generation.

Disentangled representation learning in multimodal data traces back to seminal works in Variational Autoencoders and Generative Adversarial Networks (Lee & Pavlovic, 2021; Daunhawer et al., 2021; Denton et al., 2017; Gonzalez-Garcia et al., 2018), aiming to isolate the underlying factors of variation within the data. Recent studies have increasingly explored self-supervised learning methods to capture both shared and modality-specific information (Liu et al., 2024b; Li et al., 2024; Zhang et al., 2024). In particular, information-theoretic approaches model modality-specific information as the complement of shared information, learning these representations either sequentially (Sanchez et al., 2020) or jointly (Liang et al., 2024; Pan et al., 2021). However, none of these methods provide rigorous definitions or theoretical guarantees on the optimality of disentangled representations, particularly under the multimodal scenario where the *Minimum Necessary Information* (MNI) point between the two modalities, as introduced in Fischer (2020), is often unattainable. In simple terms, the MNI criterion characterizes $Z$ as a representation of $X^1$ that perfectly captures all the important information needed to understand $X^2$, without containing any unnecessary details. Essentially, knowing $Z$ gives you the same understanding of $X^2$ as directly knowing $X^1$, and vice versa[1]. However, the shared and modality-specific information is deeply entangled in a wide range of real-world applications, i.e., the shared and modality-specific components are intertwined and result in similar observations, leading to unattainable MNI. Figure 1 illustrates an example in the context of high-content drug screens. Specifically, the two modalities capture distinct but related aspects of drug mechanism and cancer cell viability, and extracting precise shared features from each modality is not feasible. We provide additional examples of unattainable MNI in Appendix A.

In this work, we propose **Disentangled Self Supervised Learning** (DISENTANGLEDSSL), a self-supervised representation learning approach for multimodal data that effectively separates shared and modality-specific information. Building on principles of information theory, we devise a step-by-step optimization strategy to learn these representations and maximize the variational lower bound on our proposed objectives during model training. Unlike existing works, we specifically address the challenging and frequently encountered scenarios where MNI is unattainable. We further articulate a formal analysis of the optimality of each disentangled representation and offer theoretical guarantees for our algorithm's ability to achieve optimal disentanglement regardless of the attainability of the MNI point. To the best of our knowledge, this establishes the first set of analyses on the disentangled representations under such settings. Empirically, we demonstrate that DISENTANGLEDSSL successfully achieves both distinct coverage and disentanglement for representations on a suite of synthetic datasets and multiple real-world multimodal datasets. It consistently outperforms baselines on prediction tasks in the multimodal benchmarks proposed in Liang et al. (2021), as well as molecule-phenotype retrieval tasks in high-content drug screens

---

[1]Formally, $I(Z; X^1) = I(Z; X^2) = I(X^1; X^2)$ (Fischer, 2020), which we further explain in Section 2.2.1.

datasets: LINCS gene expression profiles (Subramanian et al., 2017) and RXRX19a cell imaging profiles (Cuccarese et al., 2020). To summarize, the main contributions of our work are:

- We propose DISENTANGLEDSSL, an information-theoretic framework for learning the disentangled shared and modality-specific representations of multimodal data.

- We present a comprehensive theoretical framework to study the quality of disentanglement that generalizes to settings when the Minimum Necessary Information (MNI) is unattainable. We prove that DISENTANGLEDSSL is guaranteed to learn the optimal disentangled representations.

- Empirically, we demonstrate the efficacy of DISENTANGLEDSSL across diverse synthetic and real-world multimodal datasets and tasks, including prediction tasks for vision-language data and molecule-phenotype retrieval tasks for biological data.

## 2 METHOD

In this section, we detail our proposed method, DISENTANGLEDSSL, for learning disentangled representations in multimodal data. We begin by outlining the graphical model that formalizes the problem in Section 2.1 and defining the key properties of "desirable" representations in Section 2.2. Subsequently, we describe the DISENTANGLEDSSL framework in Section 2.3 and provide theoretical guarantees on the optimality of the learned representations. Finally, in Section 2.4, we present the tractable training objectives that facilitate efficient learning for each term of our method.

### 2.1 MULTIMODAL REPRESENTATION LEARNING WITH DISENTANGLED LATENT SPACE

DISENTANGLEDSSL learns disentangled representations in latent space, separating modality-specific information from shared factors across paired observations $(X^1, X^2)$. This generative process is modeled in Figure 2. Each observation is generated from two distinct latent representations: the modality-specific representations ($Z_s^1$ and $Z_s^2$) that contain information exclusive to their respective modalities, and a shared representation ($Z_c$) that contains information common to both modalities. We refer to these as the true latents.

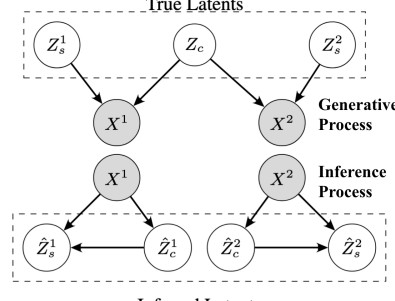

Figure 2: Graphical model.

DISENTANGLEDSSL infers the shared representation from both modalities independently, i.e. $\hat{Z}_c^1 \sim p(\cdot|X^1)$ and $\hat{Z}_c^2 \sim p(\cdot|X^2)$. The modality-specific information for each modality is encoded by variables $\hat{Z}_s^1$ and $\hat{Z}_s^2$. Note that for the true latents, $Z_s^1$ and $Z_c$ are conditionally dependent on $X^1$ due to the V-structure in the graphical model. To preserve such dependencies in the inferred latents, $\hat{Z}_s^1$ and $\hat{Z}_s^2$ are conditioned on both the respective observations and the inferred shared representations, with $\hat{Z}_s^1 \sim p(\cdot|X^1, \hat{Z}_c^1)$ and $\hat{Z}_s^2 \sim p(\cdot|X^2, \hat{Z}_c^2)$.

### 2.2 INFORMATION CRITERIA FOR THE OPTIMAL INFERRED REPRESENTATIONS

We establish information-theoretic criteria to ensure the shared and modality-specific representations are informative and disentangled, capturing key features while minimizing redundancy.

#### 2.2.1 INFORMATION BOTTLENECK PRINCIPLE AND MINIMUM NECESSARY INFORMATION

The shared representations $\hat{Z}_c^1$ and $\hat{Z}_c^2$ should effectively balance compactness and expressivity, as studied by the information bottleneck (IB) principle in both supervised and self-supervised settings (Tishby et al., 2000; Shwartz-Ziv & Tishby, 2017; Shwartz-Ziv & LeCun, 2023). The IB objective seeks to optimize the representation $Z^1$ of an observation $X^1$ in relation to a target variable $X^2$, following the Markov chain $Z^1 \leftarrow X^1 \leftrightarrow X^2$. It balances the trade-off between preserving relevant information about $X^2$, i.e. $I(Z^1; X^2)$, and compressing the representation, i.e. $I(Z^1; X^1)$ (see more details in Appendix B). The optimal representation should be both **sufficient**, i.e. $I(Z^1; X^2) = I(X^1; X^2)$ (Achille & Soatto, 2018; Shwartz-Ziv & LeCun, 2023), and **minimal** (Achille & Soatto, 2018). Based on these criteria, $Z^1$ is said to capture **Minimum Necessary**

**Information (MNI)** between $X^1$ and $X^2$ if the following holds (Fischer, 2020):

$$I(X^1; X^2) = I(Z^1; X^2) = I(Z^1; X^1)$$

MNI characterizes an ideal scenario between $X^1$ and $X^2$, where $Z^1$ captures complete information about $X^2$ with no extraneous information, i.e. $I(Z^1; X^1|X^2) = 0$[2]. It may not be attainable for an arbitrary joint distribution $p(X^1, X^2)$ (see Appendix E), particularly in general multimodal self-supervised settings (e.g. Figure 1 and Appendix A). Despite its significance, a comprehensive discussion of representation optimality when MNI is unattainable is often overlooked in prior work.

### 2.2.2 OPTIMAL SHARED REPRESENTATIONS: MNI ATTAINABLE OR NOT

We propose a definition of optimality for the shared representations that applies to both scenarios – when MNI is attainable or not, as defined in Equation (1)[3]:

$$\hat{Z}_c^{1*} = \arg\min_{Z^1} I(Z^1; X^1|X^2), \text{ s.t. } I(X^1; X^2) - I(Z^1; X^2) \leq \delta_c$$
$$\hat{Z}_c^{2*} = \arg\min_{Z^2} I(Z^2; X^2|X^1), \text{ s.t. } I(X^1; X^2) - I(Z^2; X^1) \leq \delta_c \tag{1}$$

Formally, minimizing the conditional mutual information, $I(Z^1; X^1|X^2)$, ensures that the shared representation captures only the information that is truly common to both $X^1$ and $X^2$, while discarding modality-specific details unique to $X^1$. Compared with $I(Z^1; X^1)$ in IB, it provides a more precise measure of compression and a more robust objective.

The constraint $I(X^1; X^2) - I(Z^1; X^2) \leq \delta_c$ ensures that $\hat{Z}_c^{1*}$ retains a substantial portion of the shared information between $X^1$ and $X^2$, controlling the difference within the limit $\delta_c$ and preventing significant information loss. We utilize the **IB curve** $F(\delta)$[4] (Kolchinsky et al., 2018; Gilad-Bachrach et al., 2003), representing the maximum $I(Z^1; X^2)$ for a given compression level $I(X^1; Z^1) \leq \delta$, to illustrate the optimality in Figure 3. MNI is depicted as point A, and $\hat{Z}_c^{1*}$ corresponding to $\delta_c$ is shown as point C. When MNI is attainable, setting $\delta_c = 0$ achieves MNI.

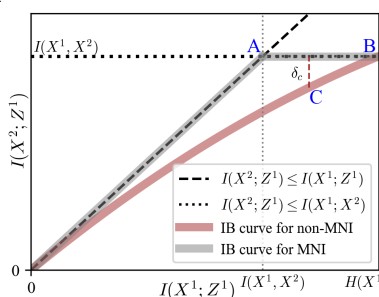

Figure 3: IB Curve

In contrast, Achille & Soatto (2018) formulated the optimization as $Z^1 = \arg\min_{Z^1: Z^1\text{-}X^1\text{-}X^2} I(X^1; Z^1), \text{ s.t. } I(Z^1; X^2) = I(X^1; X^2)$, leading to MNI when attainable. This holds in supervised settings, assuming the data label $X^2$ is a deterministic function of $X^1$, as used by previous methods (Kolchinsky et al., 2018; Fischer, 2020; Pan et al., 2021). However, in general multimodal self-supervised scenarios where MNI is not attainable, this results in point B in Figure 3, which includes information of $X^1$ that has little relevance to $X^2$ to satisfy the equality constraint, causing a gap between the objective and the ideal representation.

### 2.2.3 OPTIMAL SPECIFIC REPRESENTATIONS: ENSURING COVERAGE AND DISENTANGLEMENT

Optimal modality-specific representations, $\hat{Z}_s^1$ and $\hat{Z}_s^2$, should capture information unique to each modality, being highly informative while minimizing redundancy with the shared representations. We hence define them via the optimization problems:

$$\hat{Z}_s^{1*} = \arg\max_{Z^1} I(Z^1; X^1|X^2), \text{ s.t. } I(Z^1; \hat{Z}_c^{1*}) \leq \delta_s$$
$$\hat{Z}_s^{2*} = \arg\max_{Z^2} I(Z^2; X^2|X^1), \text{ s.t. } I(Z^2; \hat{Z}_c^{2*}) \leq \delta_s \tag{2}$$

Take $\hat{Z}_s^{1*}$ as an example. By maximizing the conditional mutual information, $I(Z^1; X^1|X^2)$, we seek to extract the unique information inherent to modality $X^1$ that is not redundant with what is

---

[2] $I(Z^1; X^1|X^2) = I(Z^1; X^1, X^2) - I(Z^1; X^2) = I(Z^1; X^1) - I(Z^1; X^2) = 0$, where the first equality is due to the property of conditional mutual information, and the second is due to the Markov structure $Z^1\text{-}X^1\text{-}X^2$.

[3] $\delta_c$ denotes a small tolerance to account for non-attainability of MNI for the modality-specific representation.

[4] The IB curve is concave (Gilad-Bachrach et al., 2003), monotonically non-decreasing, and upper bounded by line $I(Z^1; X^2) = I(Z^1; X^1)$ and line $I(Z^1; X^2) = I(X^1; X^2)$ (see details and proofs in Appendix C).

shared with $X^2$. Note that the same term is minimized in the shared representations as described in Equation 1. Integrating both the shared and the modality-specific representations together, the optimization objectives ensure full coverage of the entire information spectrum of each modality.

Meanwhile, the constraint $I(Z^1; \hat{Z}_c^{1*}) \leq \delta_s$ minimizes the overlap between modality-specific and shared representations, enhancing disentanglement by effectively separating the unique aspects of $X^1$ from the components shared with $X^2$. The parameter $\delta_s$ controls the trade-off between coverage and disentanglement.

## 2.3 DISENTANGLEDSSL: A STEP-BY-STEP OPTIMIZATION ALGORITHM

To achieve the optimal representations discussed in Section 2.2, we introduce a two-step training procedure. The first step focuses on optimizing the shared latent representation, ensuring it captures the minimum necessary information as close as possible. Building upon this, the second step utilizes the learned shared representations in step 1 to facilitate the learning of modality-specific representations. This sequential approach is formalized in the optimization objectives given in Equations 3 and 4, with the pseudocode provided in Appendix I:

**Step 1:** Learn the shared latent representations by encouraging the shared representation encoded from one modality to be highly informative about the other modality, while minimizing redundancy.

$$
\begin{aligned}
\hat{Z}_c^{1*} &= \arg\max_{Z^1} L_c^1 = \arg\max_{Z^1} I(Z^1; X^2) - \beta \cdot I(Z^1; X^1 | X^2) \\
\hat{Z}_c^{2*} &= \arg\max_{Z^2} L_c^2 = \arg\max_{Z^2} I(Z^2; X^1) - \beta \cdot I(Z^2; X^2 | X^1)
\end{aligned}
\tag{3}
$$

**Step 2:** Learn the modality-specific latent representations based on the learned shared representations from step 1.

$$
\begin{aligned}
\hat{Z}_s^{1*} &= \arg\max_{Z^1} L_s^1 = \arg\max_{Z^1} I(Z^1, \hat{Z}_c^{2*}; X^1) - \lambda \cdot I(Z^1; \hat{Z}_c^{1*}) \\
\hat{Z}_s^{2*} &= \arg\max_{Z^2} L_s^2 = \arg\max_{Z^2} I(Z^2, \hat{Z}_c^{1*}; X^2) - \lambda \cdot I(Z^2; \hat{Z}_c^{2*})
\end{aligned}
\tag{4}
$$

The hyperparameters $\beta$ and $\lambda$ control the trade-off between relevance and redundancy for the shared and modality-specific representations respectively. We use the same values of $\beta$ and $\lambda$ for both modalities since they operate on similar information scales. Our sequential training approach, instead of a joint one, stems from the self-sufficient nature of each optimization procedure where one sub-optimal representation does not enhance the learning of the other. We offer a comprehensive analysis of the optimality guarantees of this step-by-step method as follows.

### 2.3.1 OPTIMALITY GUARANTEE FOR THE LEARNED SHARED REPRESENTATIONS

This section explores how the step 1 objective, $L_c^1$, optimizes the shared representation between modalities by balancing expressivity and redundancy. We discuss its effectiveness in both scenarios–when MNI is attainable or not.

$L_c^1$ seeks representation $Z^1$ that maximizes the information shared between the modalities, i.e. $I(Z^1; X^2)$, while minimizing the information unique to each modality, i.e. $I(Z^1; X^1 | X^2)$, to capture only the essential shared content. This aligns with the conditional entropy bottleneck (CEB) objective (Fischer, 2020)[5], an extension of the IB Lagrangian $L = I(Z^1; X^2) - \tilde{\beta} I(Z^1; X^1)$ (see details in Appendix B). While it serves as a robust objective for learning shared information, its optimality remains underexplored in Fischer (2020), particularly when MNI is unattainable. Additionally, Fischer (2020) focuses on the supervised scenario where $X^2$ is the label of $X^1$, whereas we address the multimodal case with $X^1$ and $X^2$ being two data modalities, demonstrating its effectiveness in both attainable and unattainable MNI scenarios.

**When MNI is attainable**, Proposition 1 states that the step 1 optimization achieves MNI for any positive $\beta$. The proof is given in Appendix D.

---

[5] Note that $L_c^1$ is the Lagrangian formulation of the constraint optimization in Equation 1, among which the term $I(X^1; X^2)$ is irrelevant to the optimization target $Z_1$ and omitted from the objective.

**Proposition 1.** *If MNI is attainable for random variable $X^1$ and $X^2$, maximizing $L_c^1 = I(Z^1; X^2) - \beta I(Z^1; X^1|X^2)$ achieves MNI for any $\beta > 0$, i.e. $I(\hat{Z}_c^{1*}; X^1) = I(\hat{Z}_c^{1*}; X^2) = I(X^1; X^2)$, where $\hat{Z}_c^{1*} := \arg\max_{Z^1 - X^1 - X^2} L_c^1$.*

**When MNI is unattainable**, the inequalities $I(Z^1; X^1) \geq I(Z^1; X^2)$ and $I(X^1; X^2) \geq I(Z^1; X^2)$ cannot achieve equality simultaneously. This indicates an inherent trade-off between capturing the entire shared information and avoiding the inclusion of modality-specific details. More precisely, under the condition of strict concavity of the $I(Z^1; X^2) - I(Z^1; X^1)$ information curve[6], such a trade-off is presented in Proposition 2. The proof is given in Appendix D.

**Proposition 2.** *For random variables $X^1$ and $X^2$, when the IB curve $I(Z^1; X^2) = F(I(Z^1; X^1))$ is strictly concave,*
*1) there exists a bijective mapping from $\beta$ in $L_c^1$ to the value of information constraint $\delta_c$ in the definition of optimal shared latent $\hat{Z}_c^{1*}$ in Equation 1;*
*2) $\frac{\partial I(Z_\beta^{1*}; X^1)}{\partial \beta} < 0$, $\frac{\partial I(Z_\beta^{1*}; X^2)}{\partial \beta} < 0$, where $Z_\beta^{1*}$ is the optimal solution corresponding to a certain $\beta$.*

The second property implies that as $\beta$ increases, the learned representation becomes less informative and redundant, demonstrating the trade-off between expressivity and redundancy when MNI is unattainable. Furthermore, the bijection mapping between $\beta$ and $\delta_c$ allows our step 1 optimization to navigate the information frontier across various $\beta$ values, facilitating soft control and the segmentation of shared information into different degrees of granularity.

### 2.3.2 OPTIMALITY GUARANTEE FOR THE LEARNED MODALITY-SPECIFIC REPRESENTATIONS

In this section, we demonstrate the step 2 objective, $L_s^1$, ensures optimal coverage and disentanglement by showing its equivalence (or nearly equivalence) to the Lagrangian of Equation 2.

$L_s^1$ in Equation 4 learns the modality-specific representation $Z^1$ based on the optimal shared representations $\hat{Z}_c^{1*}$ and $\hat{Z}_c^{2*}$ learned from step 1. It maximizes the information coverage of the data $X^1$ through the combination of $Z^1$ and $\hat{Z}_c^{2*}$, i.e. $I(Z^1, \hat{Z}_c^{2*}; X^1)$. Simultaneously, it promotes disentanglement by limiting overlap with the shared representation $\hat{Z}_c^{1*}$ of the same modality, indicated by $I(Z^1; \hat{Z}_c^{1*})$. This objective is a Lagrangian formulation of the constraint optimization in Equation 2, however replacing $I(Z^1; X^1|X^2)$ with $I(Z^1, \hat{Z}_c^{2*}; X^1)$.

**When MNI is attainable**, as justified in Proposition 3, this substitution results in an equivalent objective, allowing $L_s^1$ to achieve optimal modality-specific representations (proof in Appendix D).

**Proposition 3.** *If MNI is attainable for random variables $X^1$ and $X^2$,*

$$\arg\max_{Z^1 - X^1 - X^2} I(Z^1; X^1|X^2) = \arg\max_{Z^1 - X^1 - X^2} I(Z^1, \hat{Z}_c^{2*}; X^1)$$

*where $\hat{Z}_c^{2*}$ is the representation of $X^2$ that satisfies MNI, i.e. $I(\hat{Z}_c^{2*}; X^1) = I(\hat{Z}_c^{2*}; X^2) = I(X^1; X^2)$.*

**When MNI is unattainable**, Proposition 4 shows such substitution yields an almost equivalent objective, subject to the value of $\delta_c$ that corresponds to the $\beta$ used in step 1 optimization. Thus, optimizing $L_s^1$ results in nearly optimal modality-specific representations (proof in Appendix D).

**Proposition 4.** *For random variables $X^1$ and $X^2$,*

$$0 \leq I(Z^1, X^2; X^1) - I(Z^1, \hat{Z}_c^{2*}; X^1) \leq \delta_c$$

*where $\hat{Z}_c^{2*}$ is the optimal representation of $X^2$ with respect to $\delta_c$ as defined in Equation 1, i.e. $\hat{Z}_c^{2*} = \arg\min_{Z^2} I(Z^2; X^2|X^1)$, s.t. $I(X^1; X^2) - I(Z^2; X^1) \leq \delta_c$.*

### 2.4 TRACTABLE TRAINING OBJECTIVES

Four terms are involved in DISENTANGLEDSSL, including maximizing the mutual information term $I(Z^1; X^2)$ and minimizing the conditional mutual information term $I(Z^1; X^1|X^2)$ for the shared

---

[6]When the IB curve is not strictly concave (i.e. partially linear), the properties hold except the linear sections. In these cases, methods like squared IB (Kolchinsky et al., 2018) can be used to achieve bijection mapping.

representations in step 1, as well as maximizing the joint mutual information term $I(Z^1, \hat{Z}_c^{2*}; X^1)$ and minimizing the mutual information term $I(Z^1; \hat{Z}_c^{1*})$ for the modality-specific representations in step 2. In this section, we introduce the tractable training objectives for each term, with detailed formulations described in Appendix G.

For the inferred shared representations, we model the distributions $\hat{Z}_c^1 \sim p(\cdot|X^1)$ and $\hat{Z}_c^2 \sim p(\cdot|X^2)$ with neural network encoders. Following the common practice in Radford et al. (2021), we use the InfoNCE objective (Oord et al., 2018) as an estimation of the mutual information term $I(Z^1; X^2)$ in $L_c^1$. We implement the conditional mutual information term $I(Z^1; X^1|X^2)$ in $L_c^1$ using an upper bound developed in Federici et al. (2019), i.e. $I(Z^1; X^1|X^2) \leq D_{\text{KL}}(p(Z^1|X^1)||p(Z^2|X^2))$. While the conditional distributions of representations are modeled as the Gaussian distribution in Federici et al. (2019), we instead use the von Mises-Fisher (vMF) distribution for $p(Z^1|X^1)$ and $p(Z^2|X^2)$ to better align with the InfoNCE objective where the representations lie on the sphere space. Specifically, $\hat{Z}_c^1 \sim \text{vMF}(\mu(X^1), \kappa), \hat{Z}_c^2 \sim \text{vMF}(\mu(X^2), \kappa)$ where $\kappa$ is a hyperparameter controlling for the uncertainty of the representations. Leveraging the formulation of the KL divergence between two vMF distributions, the training objective of $L_c^1$ is to maximize:

$$L_c^1 = -L_{\text{InfoNCE}}^c + \beta \cdot \mathbb{E}_{x^1, x^2}\left[\mu(x^1)^\top \mu(x^2)\right]$$

The inferred modality-specific representations are encoded as functions that takes both data observations and the shared representations learned in step 1 as input to account for the dependence structure illustrated in Figure 2. The term $I(Z^1, \hat{Z}_c^{2*}; X^1)$ in $L_s^1$ is optimized with the InfoNCE loss, where random augmentations of the data $X^1$ form the two views. For the mutual information term between representations, $I(Z^1; \hat{Z}_c^{1*})$ in $L_s^1$, we implement it as an orthogonal loss to encourage the marginal independence between the shared and modality-specific representations, where the marginal distribution is approximated across a training batch.

## 3 EXPERIMENTS

We conduct a simulation study and two real-world multimodal experiments to evaluate the efficacy of our proposed DISENTANGLEDSSL. We compare DISENTANGLEDSSL against a disentangled variational autoencoder baseline, DMVAE (Lee & Pavlovic, 2021), as well as various multimodal contrastive learning methods, including CLIP (Radford et al., 2021), which aligns different modalities to learn the shared representations, and FOCAL (Liu et al., 2024b), FactorCL (Liang et al., 2024) and SimMMDG (Dong et al., 2023), which capture both shared and modality-specific information. Additionally, we evaluate a joint optimization variant of DISENTANGLEDSSL, JointOpt (Pan et al., 2021), to highlight the advantages of our step-by-step approach.

### 3.1 SIMULATION STUDY

**Synthetic data generation.** We generate synthetic data based on the graphical model in Figure 2. We sample $d$-dimensional true latents $Z_s^1, Z_s^2$, and $Z_c$ independently from $\mathcal{N}(\mu_d, \Sigma_d^2)$. Using fixed transformations $T_1$ and $T_2$, we create $X^1 = T_1 \cdot [Z_s^1, Z_c]$ and $X^2 = T_2 \cdot [Z_s^2, Z_c]$. To simulate unattainable MNI, we add Gaussian noise to ensure the distribution has full support. Binary labels $Y_s^1, Y_s^2$, and $Y_c$ are constructed from the true latents and used to evaluate the information content of learned representations via linear probing accuracy. Denote $\hat{Z}_c$ as the combination of the learned shared representations of $X^1$ and $X^2$, i.e. $\hat{Z}_c = [\hat{Z}_c^1, \hat{Z}_c^2]$. Ideally, $\hat{Z}_c$ should achieve high accuracy on $Y_c$ and low on $Y_s^1$ and $Y_s^2$, while the modality-specific representations $\hat{Z}_s^1$ and $\hat{Z}_s^2$ should show the opposite pattern. Additional details on experimental settings can be found in Appendix H.1.

**Results.** We assess the performance of the learned shared and modality-specific representations for different values of $\beta$ and $\lambda$, as shown in Figure 4. For comparison, we also evaluate JointOpt, DMVAE, FOCAL, and SimMMDG across different hyperparameter settings. Specifically, for JointOpt, we vary both $a$ and $\lambda$, where $a$ controls the joint mutual information terms $I(\hat{Z}_s^1, \hat{Z}_c^2; X^1)$ and $I(\hat{Z}_s^2, \hat{Z}_c^1; X^2)$, and $\lambda$ adjusts the mutual information term between representations, similar to DISENTANGLEDSSL.

Figure 4a illustrates the performance of shared representation $\hat{Z}_c$ learned by DISENTANGLEDSSL across different values of $\beta$. For lower values of $\beta$, $\hat{Z}_c$ captures both shared and specific features, as indicated by linear probing accuracy on $Y_s^1, Y_s^2$, and $Y_c$ exceeding 0.5. As $\beta$ increases, all accuracies

decrease, reflecting the trade-off between expressivity and redundancy controlled by $\delta_c$ when MNI is unattainable. This trend aligns well with Figure 3 and Proposition 2. A comparison of the performance of the shared representation with baseline methods is provided in Appendix H.1. We further examine such trade-off on synthetic data with varying levels of multimodal entanglement, where some dimensions are aligned across modalities with MNI attainable and others remain entangled with MNI unattainable. Detailed results are provided in Appendix H.1.

Given the shared representations $\hat{Z}_c^1$ and $\hat{Z}_c^2$ learned in step 1 for a fixed $\beta$, we then learn the corresponding modality-specific representations $\hat{Z}_s^1$ and $\hat{Z}_s^2$ with varying $\lambda$. Figure 4b shows the performance of DISENTANGLEDSSL in contrast to other baseline methods, where dots are connected according to a descending order of their corresponding $\lambda$ values[7]. The ideal modality-specific representation $\hat{Z}_s^1$ should maximize unique information from $X^1$, shown by high accuracy on $Y_s^1$, while minimizing shared information with $X^2$, indicated by low accuracy on $Y_c$. Therefore, a bottom-right point is preferred in Figure 4b. As illustrated in Figure 4b, DISENTANGLEDSSL outperforms all other methods across various hyperparameter settings, especially JointOpt, demonstrating the effectiveness of the stepwise optimization procedure. Results on $\hat{Z}_s^2$ are provided in Appendix H.1.

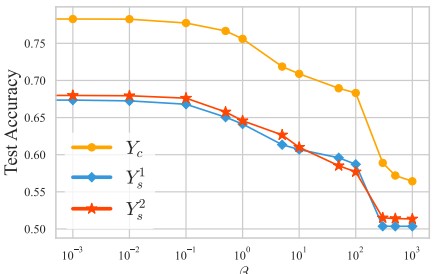
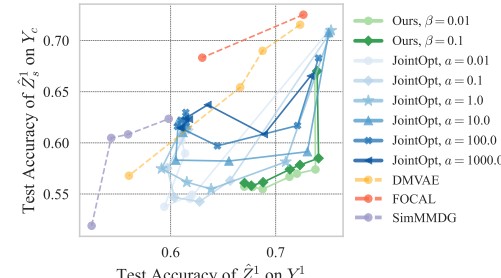

(a) Performance of shared representation $\hat{Z}_c$ learned with different values of $\beta$.

(b) Performance of modality-specific representation $\hat{Z}_s^1$ learned with different $\beta$ and $\lambda$, in comparison to baselines.

Figure 4: Simulation study results.

## 3.2 MULTIBENCH

**Dataset description.** We utilize the real-world multimodal benchmark from MultiBench (Liang et al., 2021), which includes curated datasets across various modalities, such as text, images, and tabular data, along with a downstream label that we expect shared and specific information to have varying importance for. These datasets cover a variety of domains including healthcare, affective computing and multimedia research areas. We follow the same setting (dataset splitting, encoder architecture, pre-extracted features) as in Liang et al. (2024). Additional details can be found in Appendix H.2.

**Results.** We evaluate the linear probing accuracy of representations learned by each pretrained model, as shown in Table 1. FactorCL-emb refers to the embeddings of FactorCL before projection heads, while FactorCL-proj uses the concatenation of all projection head outputs. All models use representations (or concatenation of them, if applicable) with the same dimensionality. We present the best results among three representations–shared, modality-specific, and the concatenation of both–for all methods in Table 1, with results for each representation in Ap-

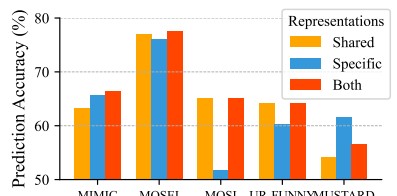

Figure 5: Results of different representations learned by DISENTANGLEDSSL.

pendix H.2. DISENTANGLEDSSL consistently outperforms baselines across datasets, demonstrating its ability to capture valuable information for downstream tasks. Further, we show the performance of each representation learned through DISENTANGLEDSSL in Figure 5. Combining shared and specific representations of DISENTANGLEDSSL improves performance in most cases, showing that both

---

[7]For FOCAL, we tune the hyperparameters $a$ and $\lambda$, defined similarly to JointOpt, where $a$ controls the terms $I(\hat{Z}_s^1; X^1)$ and $I(\hat{Z}_s^2; X^2)$ and $\lambda$ adjusts the orthogonal loss between shared and specific representations. For DMVAE, we tune $\lambda$ which denotes the weight of the KL divergence term. We show the best-performing results of FOCAL and DMVAE across hyperparameters in Figure 4b, with full results available in Appendix H.1.

contribute to label prediction, while in MOSI and MUSTARD, only shared or specific information contributes significantly. For example, for MUSTARD, specific representations capture nuances of sarcasm, such as sardonic expressions or ironic tone, which are crucial for sarcasm prediction.

Table 1: Prediction accuracy (%) of the representations learned by different methods on MultiBench datasets and standard deviations over 3 random seeds.

| Dataset | MIMIC | MOSEI | MOSI | UR-FUNNY | MUSTARD |
|---------|-------|-------|------|----------|---------|
| CLIP | 64.97 (0.60) | 76.87 (0.45) | 64.24 (0.88) | 62.73 (0.92) | 56.04 (4.19) |
| FactorCL-emb | 65.25 (0.45) | 71.80 (0.64) | 62.97 (0.81) | 63.29 (2.07) | 56.76 (4.66) |
| FactorCL-proj | 59.43 (1.70) | 74.61 (1.65) | 56.02 (1.26) | 61.25 (0.47) | 55.80 (2.18) |
| FOCAL | 64.42 (0.34) | 76.77 (0.51) | 63.65 (1.09) | 63.17 (0.96) | 58.21 (2.21) |
| JointOpt | 66.11 (0.64) | 76.71 (0.14) | 65.02 (1.96) | 63.58 (1.45) | 57.73 (4.12) |
| DISENTANGLEDSSL | **66.44** (0.31) | **77.45** (0.06) | **65.16** (0.81) | **64.24** (1.54) | **61.60** (2.61) |

## 3.3 HIGH-CONTENT DRUG SCREENING

**Dataset description.** As characterized in Figure 1, we use two high-content drug screening datasets which provide phenotypic profiles after drug perturbation: RXRX19a (Subramanian et al., 2017) containing cell imaging profiles, and LINCS (Cuccarese et al., 2020) containing L1000 gene expression profiles. For RXRX19a, we select data from the HRCE cell line under active SARS-CoV-2 condition, totaling 1,661 drugs and 10,117 profiles ($\sim$ 6 replicates per drug). Hand-crafted image features from CellProfiler (McQuin et al., 2018) are used as encoder inputs. For LINCS, we use the data curated in Wang et al. (2023a), selecting drugs with full observations across all 9 core cell lines and randomly sampling 3 replicates per drug to minimize selection bias, resulting in 3,315 drugs and 86,833 profiles. We conduct train-validation-test splitting according to molecules. Models are pretrained to learn representations of molecular structures and their corresponding phenotypes. We provide additional details on experimental settings in Appendix H.3.

Table 2: Retrieving acccuracy and mean reciprocal rank (MRR) of molecule-phenotype retrieval.

| Dataset | Top N Acc (%) | N=1 | N=5 | N=10 | N=20 | N=30 | MRR |
|---------|---------------|-----|-----|------|------|------|-----|
| RXRX19a | Random | 0.30 | 1.50 | 3.00 | 6.01 | 9.01 | - |
| | CLIP | 3.30(0.40) | 8.33(0.52) | 11.59(0.20) | 16.38(0.56) | 20.22(0.67) | 0.103(0.001) |
| | DMVAE | 3.85(0.36) | 8.76(0.30) | 11.84(0.32) | 16.20(0.83) | 20.17(0.85) | 0.106(0.002) |
| | JointOpt | 3.41(0.49) | 8.54(0.14) | 11.64(0.14) | 16.79(0.27) | 20.71(1.11) | 0.110(0.002) |
| | FOCAL | 3.61(0.51) | 8.71(0.69) | 11.94(0.74) | 16.86(0.62) | 20.81(1.07) | 0.108(0.003) |
| | DISENTANGLEDSSL ($\beta = 0$) | 3.39(0.54) | 8.25(0.33) | 11.53(0.20) | 16.28(0.10) | 20.20(0.45) | 0.109(0.001) |
| | DISENTANGLEDSSL | **4.03**(0.39) | **9.62**(0.20) | **13.12**(0.23) | **18.36**(0.42) | **22.82**(0.45) | **0.111**(0.001) |
| LINCS | Random | 0.15 | 0.75 | 1.51 | 3.02 | 4.52 | - |
| | CLIP | 3.95(0.04) | 10.81(0.06) | 15.10(0.20) | 21.08(0.39) | 29.44(0.30) | 0.146(0.001) |
| | DMVAE | 4.31(0.09) | 11.45(0.13) | 15.88(0.17) | 21.85(0.27) | 29.41(0.33) | 0.156(0.001) |
| | JointOpt | **4.67**(0.09) | 11.60(0.11) | 16.02(0.15) | 22.01(0.21) | 30.08(0.39) | 0.161(0.001) |
| | FOCAL | 4.34(0.17) | 11.24(0.19) | 15.74(0.26) | 21.48(0.08) | 29.32(0.08) | 0.157(0.002) |
| | DISENTANGLEDSSL ($\beta = 0$) | 4.36(0.13) | 11.27(0.39) | 15.81(0.47) | 21.71(0.48) | 29.65(0.54) | 0.158(0.001) |
| | DISENTANGLEDSSL | 4.48(0.21) | **11.70**(0.40) | **16.39**(0.39) | **22.68**(0.58) | **30.84**(0.66) | **0.163**(0.002) |

**Molecule-phenotype retrieval using shared representations.** We evaluate the shared representations in the molecule-phenotype retrieval task, where the goal is to identify molecules from the whole test set that are most likely to induce a specific phenotype. The shared information, which connects the molecular structure and phenotype, plays a key role in this task. We tune $\beta$ according to validation set performance and show results of top N accuracy (N=1,5,10,20,30) and mean reciprocal rank (MRR) on the test set in Table 2. DISENTANGLEDSSL consistently outperforms baselines on both datasets, effectively capturing relevant shared features while excluding irrelevant modality-specific information. Notably, compared to the variant without the information bottleneck constraint, i.e. DISENTANGLEDSSL ($\beta = 0$), the full DISENTANGLEDSSL model preserves critical shared features, achieving superior performance in this retrieval task, where shared information is essential.

**Disentanglement measurement.** To assess the effectiveness of learning modality-specific representations, we introduce the Reconstruction Gain (RG) metric, which quantifies the disentanglement and complementariness between shared and modality-specific representations. Specifically, we train 2-layer MLP decoders on the training set to reconstruct the original data from the shared and modality-specific representations, both individually and jointly, and compute the reconstruction $R^2$ on the test set for each case. A higher gain in $R^2$ when using combined representations in comparison to individual ones indicates lower redundancy and better disentanglement.

As shown in Table 6, DISEN-TANGLEDSSL achieves the highest RG scores across both modalities and datasets, demonstrating superior disentanglement. In contrast, other methods show redundancy due to insufficient constraints for disentanglement

Table 3: Reconstruction gain (RG) in $R^2$ of representations for each modality (molecular structure/phenotype).

| Dataset | RXRX19a | | LINCS | |
|---|---|---|---|---|
| Metric | RG-molecule | RG-phenotype | RG-molecule | RG-phenotype |
| FOCAL | 0.117(0.006) | 0.547(0.001) | 0.122(0.001) | 0.618(0.002) |
| DMVAE | 0.123(0.002) | 0.545(0.003) | 0.139(0.002) | 0.605(0.003) |
| JointOpt | 0.130(0.001) | 0.524(0.001) | 0.103(0.000) | 0.604(0.002) |
| DISENTANGLEDSSL | **0.153**(0.003) | **0.591**(0.001) | **0.143**(0.000) | **0.622**(0.002) |

during pretraining. We also test the representations from the pretrained model on the counterfactual generation task, with detailed results provided in Appendix H.3.

## 4 RELATED WORK

**Disentangled representations in VAEs and GANs.** Disentangled representation learning originated from works on Variational Autoencoders (VAEs) and Generative Adversarial Networks (GANs), focusing on isolating underlying data variations linked to or independent of labels (e.g., digit identity vs. writing style in MNIST) (Gonzalez-Garcia et al., 2018; Mathieu et al., 2016; Bouchacourt et al., 2018; Daunhawer et al., 2021). This concept extended to multimodal data like text and image (Lee et al., 2021) and content and pose in video (Denton et al., 2017), typically using self- and cross-reconstruction with adversarial loss to learn shared and specific components. However, these methods often address simple cases with attainable MNI, lacking a comprehensive analysis for complex real-world multimodal scenarios where MNI is not attainable.

**Information bottleneck and its variants.** The information bottleneck (IB) principle has been used to analyze and optimize deep neural networks and the learned representations from the information theory perspective in both supervised and self-supervised settings (Tishby et al., 2000; Gilad-Bachrach et al., 2003; Shwartz-Ziv & Tishby, 2017; Kolchinsky et al., 2018; Rodríguez Gálvez et al., 2020; Tsai et al., 2020; Wang et al., 2023b; Shwartz-Ziv & LeCun, 2023), with extensions like the conditional entropy bottleneck (CEB) (Fischer, 2020; Lee et al., 2021) being developed to better refine the information plane. IB has been applied to learn molecular representations from drug screening data (Liu et al., 2024a). However, these approaches primarily focus on extracting shared features while minimizing specific features, limiting their practical use. Other methods (Pan et al., 2021; Sanchez et al., 2020) learn disentangled representations by optimizing mutual information and using adversarial loss but are confined to single-modality scenarios and do not address complex multimodal settings with unattainable MNI.

**Multimodal disentanglement in self-supervised learning.** Recent studies have explored disentangled representation learning in multimodal self-supervised contexts. Liang et al. (2024) factorizes and optimizes mutual information bounds to capture the union of shared and unique features, while not penalizing for redundancy between latents. Zhang et al. (2024) learns disentangled representations of cell state for multimodal data in biological contexts. Liu et al. (2024b) and Li et al. (2024) use mutual information optimization with orthogonal/MMD regularizations for disentangling multimodal time-series signals and T-cell receptor segments, respectively. Li et al. (2022) and Dong et al. (2023) address the separation of shared and specific information, either among views of images or in the domain generalization setting. However, these methods lack a general framework for understanding information content, particularly when data modalities are deeply entangled in real-world applications.

## 5 CONCLUSIONS

We present DISENTANGLEDSSL, a self-supervised approach for learning disentangled multimodal representations that effectively separates shared and modality-specific features. Built on a comprehensive theoretical framework, DISENTANGLEDSSL addresses the challenging scenarios where the Minimum Necessary Information (MNI) is unattainable, enhancing its applications to complex real-world multimodal settings and improving interpretability and robustness. Our empirical results show that DISENTANGLEDSSL outperforms baselines across various synthetic and real-world datasets, including vision-language prediction tasks and molecule-phenotype retrieval for high-content drug screens, demonstrating its ability to learn optimal disentangled representations for diverse applications.

## 6 REPRODUCIBILITY STATEMENT

For the theoretical results presented in the paper, we provide complete proofs in Appendix D. For the experimental results, we provide detailed explanations of the algorithm implementations and experimental setup in Section 3, Appendix G, and Appendix H. For the datasets used in the experiments, we utilize publicly available datasets and follow the data processing procedures in prior works, with additional details elaborated in Section 3 and Appendix H. The code can be found in https://github.com/uhlerlab/DisentangledSSL.

## ACKNOWLEDGEMENT

XZ, ST and CU were partially supported by the Eric and Wendy Schmidt Center at the Broad Institute, NCCIH/NIH (1DP2AT012345), ONR (N00014-22-1-2116) and DOE (DE-SC0023187). SG and SJ acknowledge the support of the NSF Award CCF-2112665 (TILOS AI Institute), an Alexander von Humboldt fellowship, and Office of Naval Research grant N00014-20-1-2023 (MURI ML-SCOPE). CW and TJ acknowledge support from the Machine Learning for Pharmaceutical Discovery and Synthesis (MLPDS) consortium, the DTRA Discovery of Medical Countermeasures Against New and Emerging (DOMANE) threats program, and the NSF Expeditions grant (award 1918839) Understanding the World Through Code.

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

## A  ADDITIONAL EXAMPLES OF UNATTAINABLE MNI

In this section, we present two additional examples where the Minimum Necessary Information (MNI) is unattainable. These examples offer a clear understanding of how unattainable MNI manifests in data, providing practical insights and making the concept more accessible.

Firstly, we show an example in a simple mathematical case. As illustrated in Figure 6, we have true latent variables are independently drawn from a Bernoulli distribution, i.e. $Z_s^1, Z_c, Z_s^2 \sim$ Bernoulli$(0.5)$, and observations $X^1$ and $X^2$ are generated according to $X^1 = \text{OR}(Z_c, Z_s^1)$ and $X^2 = \text{OR}(Z_c, Z_s^2)$. In this scenario, if we observe $X^1 = 1$, we are not able to distinguish whether this result is due to $Z_c = 1$, or $Z_s^1 = 1$, or both. Thus, extracting purely shared features from the observations is infeasible.

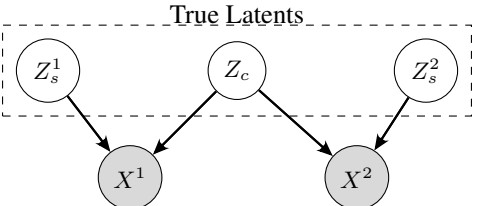

Figure 6: Example of unattainble MNI.

Further, we provide an example in 3D geometry, as illustrated in Figure 7. For information such as the height of the cone and the number of spheres, one modality conveys the complete information while the other conveys partial information. This ambiguity makes it challenging to classify the information as purely shared or modality-specific, placing it in an uncertain category.

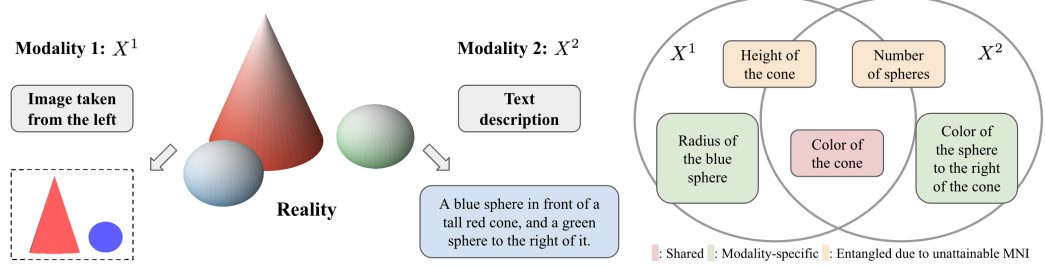

Figure 7: Image modality ($X_1$) and text modality ($X_2$) of an underlying reality. The Venn diagram illustrates shared and specific information between modalities $X_1$ and $X_2$: shared content is shown in red, modality-specific content in green, and entangled content due to unattainable MNI in orange. For example, for the cone's height, the image conveys full information, while the text (i.e. "tall red cone") provides partial information. Similarly, for the number of spheres, the text specifies it completely, whereas the image indicates there is at least one sphere.

## B  BACKGROUND ON INFORMATION BOTTLENECK

The Information Bottleneck (IB) principle provides a powerful theoretical foundation for our method. IB objective seeks to find a representation $Z^1$ of a random variable $X^1$ that optimally trades off the preservation of relevant information with compression (Tishby et al., 2000). Relevance is defined as the mutual information, $I(Z^1; X^2)$, between the representation $Z^1$ and another target variable $X^2$. Compression is enforced by constraining the mutual information between the representation and the original data, $I(X^1; Z^1)$, to lie below a specified threshold. This is can be formalize in terms of the following constrained optimization problem:

$$\arg\max_{Z^1} I(Z^1; X^2), s.t. \ I(Z^1; X^1) \le \delta \tag{5}$$

where $Z^1$ is from a set of random variables that obey the Markov chain $Z^1 \leftrightarrow X^1 \leftrightarrow X^2$. In practice, Equation 5 can be optimized by minimizing the IB Lagrangian in Equation 6.

$$\mathcal{L} = I(Z^1; X^1) - \beta I(Z^1; X^2) \tag{6}$$

Here, the Lagrangian multiplier $\beta$ controls the emphasis placed on compression versus expressiveness.

Fischer (2020) extends IB to the conditional entropy bottleneck (CEB) objective, which utilizes the conditional mutual information term $I(Z^1; X^1|X^2)$ in place of the mutual information term $I(Z^1; X^1)$ in Equation 6:

$$\mathcal{L} = I(Z^1; X^1|X^2) - \beta I(Z^1; Z^2)$$

While being equivalent to the IB Lagrangian in terms of the information criterion, CEB rectifies the information plane to precisely measure compression using the conditional mutual information, and offers a clear assessment of how close the compression is to being optimal.

Note that although CEB aligns with our step 1 objective in Equation 3, its optimality has not been fully examined in Fischer (2020), particularly under scenarios where MNI is unattainable. Furthermore, Fischer (2020) primarily addresses a supervised learning context, where $X^2$ serves as the label for $X^1$. In contrast, our study tackles the more complex multimodal setting, where $X^1$ and $X^2$ are two distinct data modalities. We extend the analysis to both attainable and unattainable MNI cases, demonstrating the efficacy of our approach in capturing shared and modality-specific information under these challenging scenarios. This broadens the applicability of CEB beyond traditional label-based supervised settings to multimodal data with complex modality entanglements.

## C  PROPERTIES OF THE IB CURVE

The **information plane** is a helpful visualization of the information bottleneck principle, which utilizes $I(X^1; Z^1)$ and $I(X^2; Z^1)$ as its coordinates that represents the trade-off between compression and prediction (Tishby et al., 2000). The frontier of all possible $Z^1$s is called the **IB curve** or $F(r)$ (Kolchinsky et al., 2018) as follows:

$$F(r) := \max_{Z^1: Z^1 - X^1 - X^2} I(Z^1; X^2) \text{ s.t. } I(X^2; Z^1) \leq r$$

The IB curve has the following properties:

- The IB curve is concave (Lemma 5 in Gilad-Bachrach et al. (2003)) and monotonically non-decreasing.
- The IB curve is upper bounded by line $I(X^2; Z^1) = I(X^1; Z^1)$ and line $I(X^2; Z^1) = I(X^1; X^2)$, according to the Markov relationship $Z^1 - X^1 - X^2$.
- When the MNI point is attainable for random variables $X^1$ and $X^2$, the IB curve has the following formula (Rodríguez Gálvez et al., 2020; Kolchinsky et al., 2018):

$$I(Z^1; X^2) = \begin{cases} I(Z^1; X^1), & \text{if } I(Z^1; X^1) \leq I(X^1; X^2) \\ I(X^1; X^2), & \text{if } I(X^1; X^2) < I(Z^1; X^1) \leq H(X^1) \end{cases}$$

and the Gateaux derivative of $I(X^2; Z^1)$ with respect to $p(z^1|x^1)$ (as used in (Pan et al., 2021)) doesn't exist at the MNI point (see details in Appendix F).

## D  PROOFS

### D.1  PROOF OF PROPOSITION 1

**Proposition 1.** *If MNI is attainable for random variable $X^1$ and $X^2$, maximizing $L_c^1 = I(Z^1; X^2) - \beta I(Z^1; X^1|X^2)$ achieves MNI for any $\beta > 0$, i.e. $I(\hat{Z}_c^{1*}; X^1) = I(\hat{Z}_c^{1*}; X^2) = I(X^1; X^2)$, where $\hat{Z}_c^{1*} := \arg\max_{Z^1 - X^1 - X^2} L_c^1$.*

*Proof.* Based on data processing inequality and the Markov relationship $\hat{Z}_c^1 \leftarrow X^1 \leftrightarrow X^2$, $I(\hat{Z}_c^1; X^2) \leq I(X^1; X^2)$. Based on the non-negativity of conditional mutual information, $I(\hat{Z}_c^1; X^1|X^2) \geq 0$. Thus, for $\beta > 0$,

$$L_c^1 = I(\hat{Z}_c^1; X^2) - \beta \cdot I(\hat{Z}_c^1; X^1|X^2) \leq I(X^1; X^2), \ \forall \hat{Z}_c^1$$

where the equality holds when $I(\hat{Z}_c^1; X^2) = I(X^1; X^2)$ and $I(\hat{Z}_c^1; X^1|X^2) = 0$

Meanwhile,

$$I(\hat{Z}_c^1; X^1|X^2) = I(\hat{Z}_c^1; X^1, X^2) - I(\hat{Z}_c^1; X^2) = I(\hat{Z}_c^1; X^1) - I(\hat{Z}_c^1; X^2)$$

where the second equality is due to the conditional independence $\hat{Z}_c^1 \perp\!\!\!\perp X^2|X^1$ in the Markov relationship.

Therefore, $L_c^1$ achieves maximality $I(X^1; X^2)$ iff $I(\hat{Z}_c^1; X^2) = I(X^1; X^2) = I(\hat{Z}_c^1; X^1)$, i.e. $\hat{Z}_c^1$ achieves MNI.

$\square$

### D.2 PROOF OF PROPOSITION 2

**Proposition 2.** *For random variables $X^1$ and $X^2$, when the IB curve $I(Z^1; X^2) = F(I(Z^1; X^1))$ is strictly concave,*
*1) there exists a bijective mapping from $\beta$ in $L_c^1$ to the value of information constraint $\delta_c$ in the definition of optimal shared latent $\hat{Z}_c^{1*}$ in Equation 1;*
*2) $\frac{\partial I(Z_\beta^{1*}; X^1)}{\partial \beta} < 0$, $\frac{\partial I(Z_\beta^{1*}; X^2)}{\partial \beta} < 0$, where $Z_\beta^{1*}$ is the optimal solution corresponding to a certain $\beta$.*

*Proof.* Since the IB curve $I(Z^1; X^2) = f_{IB}(I(Z^1; X^1))$ is monotonically non-decreasing and strictly concave, it is monotonically increasing. When $Z^1 = X^1$, $f_{IB}$ achieves the maximum point $(H(X^1), I(X^1; X^2))$. Thus, $I(X^1; X^2) \geq I(X^1; X^2) - I(\hat{Z}_c^{1*}; X^2) \geq 0$ and is monotonically decreasing. Then there is a bijective mapping between $\delta = I(X^1; X^2) - I(\hat{Z}_c^{1*}; X^2) \in [0, I(X^1; X^2)]$ and points in the IB curve.

Since $I(Z^1; X^1|X^2) = I(Z^1; X^1) - I(Z^1; X^2)$, we have

$$I(Z^1; X^1|X^2) = f_{IB}^{-1}(I(Z^1; X^2)) - I(Z^1; X^2) := f_{CIB}^{-1}(I(Z^1; X^2))$$

where $f_{CIB}(r) = (f_{IB}^{-1}(r) - r)^{-1}$. Since $f_{IB}(r) \leq r$, the equality holds when $r = 0$, and $f_{IB}$ is strictly concave, we have $\frac{df_{IB}}{dr} \leq \frac{df_{IB}}{dr}|_{r=0} < 1$. Thus $\frac{df_{IB}^{-1}}{dr} > 1$ and

$$\frac{df_{CIB}}{dr} = (\frac{df_{IB}^{-1}}{dr} - 1)^{-1} > 0$$

Furthermore, since $\frac{df_{IB}^2}{dr^2} < 0$ (strict concavity of $f_{IB}$), we have

$$\frac{df_{CIB}^2}{dr^2} = -\frac{1}{(\frac{df_{IB}^{-1}}{dr} - 1)^2} \cdot (-\frac{\frac{df_{IB}^2}{dr^2}}{(\frac{df_{IB}}{dr})^2}) < 0$$

Therefore, $f_{CIB}$ is also monotonically increasing and strictly concave. Then there is a bijective mapping between $\delta = I(X^1; X^2) - I(\hat{Z}_c^{1*}; X^2) \in [0, I(X^1; X^2)]$ and points in the CIB curve $f_{CIB}$.

Since $f_{CIB}$ is increasing, the inequality constraint can be replaced by the equality constraint, i.e.

$$\hat{Z}_c^{1*} = \underset{I(Z^1;X^2) \geq I(X^1;X^2)-\delta}{\arg\min} I(Z^1; X^1|X^2) = \underset{I(Z^1;X^2) = I(X^1;X^2)-\delta}{\arg\min} I(Z^1; X^1|X^2)$$

The corresponding Lagrangian function is

$$L = I(Z^1; X^1|X^2) - \tilde{\beta} \cdot (I(Z^1; X^2) - I(X^1; X^2) + \delta)$$
$$= f_{CIB}^{-1}(I(Z^1; X^2)) - \tilde{\beta} \cdot I(Z^1; X^2) + \tilde{\beta} \cdot (I(X^1; X^2) - \delta)$$

Based on the Lagrangian multiplier theorem, the optimal $\hat{Z}_c^{1*}$ is achieved when

$$\frac{dL}{dI(Z^1; X^2)} = \frac{df_{CIB}^{-1}(I(Z^1; X^2))}{dI(Z^1; X^2)} - \tilde{\beta} = 0$$

$$\frac{dL}{d\tilde{\beta}} = -I(Z^1; X^2) + I(X^1; X^2) - \delta = 0$$

Thus,

$$\tilde{\beta} = \frac{df^{-1}_{CIB}(I(Z^1; X^2))}{dI(Z^1; X^2)}\Big|_{I(Z^1;X^2)=I(X^1;X^2)-\delta} \tag{7}$$

i.e. $\tilde{\beta}$ is the slope of $f^{-1}_{CIB}$ when $I(Z^1; X^2) = I(X^1; X^2) - \delta$. Therefore, there is a bijective mapping between $\beta = \tilde{\beta}^{-1}$ in $L_c^1$ and points in $f_{CIB}$, and thus $\delta$.

For any $\beta_1, \beta_2 > 0$ and $\beta_1 > \beta_2$, we have $\tilde{\beta}_1 = \beta_1^{-1} < \beta_2^{-1} = \tilde{\beta}_2$. According to formula 7,

$$\tilde{\beta}_1 = \frac{df^{-1}_{CIB}(I(Z^1; X^2))}{dI(Z^1; X^2)}\Big|_{I(Z^1;X^2)=I(Z^*_{\beta_1};X^2)}, \ \tilde{\beta}_2 = \frac{df^{-1}_{CIB}(I(Z^1; X^2))}{dI(Z^1; X^2)}\Big|_{I(Z^1;X^2)=I(Z^*_{\beta_2};X^2)}$$

Since $\tilde{\beta}_1 < \tilde{\beta}_2$ and $f_{CIB}$ is a strictly concave function, $f^{-1}_{CIB}$ is strictly convex and we have $I(Z^*_{\beta_1}; X^2) < I(Z^*_{\beta_2}; X^2)$. Furthermore, since $f_{IB}$ is monotonically increasing, we have $I(Z^*_{\beta_1}; X^1) < I(Z^*_{\beta_2}; X^1)$. Therefore,

$$\frac{\partial I(Z^*_\beta; X^2)}{\partial \beta} < 0, \ \frac{\partial I(Z^*_\beta; X^1)}{\partial \beta} < 0$$

$\square$

### D.3 Proof of Proposition 3

**Proposition 3.** *If MNI is attainable for random variables $X^1$ and $X^2$,*

$$\underset{Z^1-X^1-X^2}{\arg\max} \ I(Z^1; X^1|X^2) = \underset{Z^1-X^1-X^2}{\arg\max} \ I(Z^1, \hat{Z}_c^{2*}; X^1)$$

*where $\hat{Z}_c^{2*}$ is the representation based on $X^2$ that satisfies MNI, i.e. $I(\hat{Z}_c^{2*}; X^1) = I(\hat{Z}_c^{2*}; X^2) = I(X^1; X^2)$.*

*Proof.* Based on the graphical model, $\hat{Z}_c^2 \perp\!\!\!\perp X^1|X^2, Z^1$, we have $I(\hat{Z}_c^2; X^1|Z^1, X^2) = 0$. Thus,

$$I(Z^1, X^2; X^1) = I(Z^1, X^2; X^1) + I(\hat{Z}_c^2; X^1|Z^1, X^2)$$
$$= I(Z^1, X^2, \hat{Z}_c^2; X^1) = I(Z^1, \hat{Z}_c^2; X^1) + I(X^1; X^2|Z^1, \hat{Z}_c^2) \tag{8}$$

Meanwhile,

$$I(X^1; X^2|\hat{Z}_c^2) - I(X^1; X^2|Z^1, \hat{Z}_c^2) = I(X^2; Z^1|\hat{Z}_c^2) - I(X^2; Z^1|X^1, \hat{Z}_c^2) = I(X^2; Z^1|\hat{Z}_c^2)$$

where the second equality is due to the conditional independence $X^2 \perp\!\!\!\perp Z^1|X^1, \hat{Z}_c^2$ in the graphical model.

When $\hat{Z}_c^2$ is the MNI point $\hat{Z}_c^{2*}$, $I(X^1; \hat{Z}_c^{2*}) = I(X^1; X^2)$. Thus,

$$I(X^1; X^2|\hat{Z}_c^{2*}) = I(X^1; X^2, \hat{Z}_c^{2*}) - I(X^1; \hat{Z}_c^{2*}) = I(X^1; X^2) - I(X^1; \hat{Z}_c^{2*}) = 0$$

where the first equality is due to the Markov relationship $X^1 \leftrightarrow X^2 \to \hat{Z}_c^{2*}$.

Then we have $-I(X^1; X^2|Z^1, \hat{Z}_c^{2*}) = I(X^2; Z^1|\hat{Z}_c^{2*})$. Due to the non-negativity of conditional mutual information,

$$I(X^1; X^2|Z^1, \hat{Z}_c^{2*}) = I(X^2; Z^1|\hat{Z}_c^{2*}) = 0 \tag{9}$$

Based on formula 8 and 9,

$$I(Z^1, X^2; X^1) = I(Z^1, \hat{Z}_c^{2*}; X^1)$$

Since $I(Z^1; X^1|X^2) = I(Z^1, X^2; X^1) - I(X^1; X^2)$ and $I(X^1; X^2)$ is a constant value irrelevant to $Z^1$, maximizing $I(Z^1; X^1|X^2)$ is equivalent to maximizing $I(Z^1, X^2; X^1) = I(Z^1, \hat{Z}_c^{2*}; X^1)$.

$\square$

## D.4 PROOF OF PROPOSITION 4

**Proposition 4.** *For random variables $X^1$ and $X^2$,*

$$0 \leq I(Z^1, X^2; X^1) - I(Z^1, \hat{Z}_c^{2*}; X^1) \leq \delta_c$$

*where $\hat{Z}_c^{2*}$ is the optimal representation based on $X^2$ with respect to $\delta_c$ as defined in Equation 1, i.e.*
*$\hat{Z}_c^{2*} = \arg\min_{Z^2} I(Z^2; X^2 | X^1)$, s.t. $I(X^1; X^2) - I(Z^2; X^1) \leq \delta_c$.*

*Proof.* According to formula 8 and 9,

$$I(Z^1, X^2; X^1) = I(Z^1, \hat{Z}_c^2; X^1) + I(X^1; X^2 | Z^1, \hat{Z}_c^2)$$

$$I(X^1; X^2 | Z^1, \hat{Z}_c^2) = I(X^1; X^2 | \hat{Z}_c^2) - I(Z^1; X^2 | \hat{Z}_c^2)$$

Since $\hat{Z}_c^{2*}$ is the optimal latent under $\delta$, $I(X^1; X^2) - I(\hat{Z}_c^{2*}; X^1) \leq \delta$. Thus,

$$I(X^1; X^2 | \hat{Z}_c^{2*}) = I(X^1; X^2) - I(X^1; \hat{Z}_c^{2*}) \leq \delta$$

Meanwhile, $I(Z^1; X^2 | \hat{Z}_c^2) \geq 0$. Then we have $I(X^1; X^2 | Z^1, \hat{Z}_c^2) \leq \delta - I(Z^1; X^2 | \hat{Z}_c^2) \leq \delta$. Therefore,
$$0 \leq I(Z^1, X^2; X^1) - I(Z^1, \hat{Z}_c^2; X^1) = I(X^1; X^2 | Z^1, \hat{Z}_c^2) \leq \delta$$

i.e. $|I(Z^1, X^2; X^1) - I(Z^1, \hat{Z}_c^2; X^1)| \leq \delta$

Since $I(Z^1; X^1 | X^2) = I(Z^1, X^2; X^1) - I(X^1; X^2)$ and $I(X^1; X^2)$ is a constant value irrelevant to $Z^1$, maximizing $I(Z^1; X^1 | X^2)$ is equivalent to maximizing $I(Z^1, X^2; X^1) \in [I(Z^1, \hat{Z}_c^{2*}; X^1), I(Z^1, \hat{Z}_c^{2*}; X^1) + \delta]$.

$\square$

# E SUFFICIENT CONDITIONS FOR MNI

In this section, we explore the conditions for both MNI attainable and unattainable cases. Here, we use $X$ and $Y$ to represent the two modalities, rather than $X^1$ and $X^2$ in other sections. Although prior works have not provided a comprehensive necessary and sufficient condition for the attainability of MNI, and such analysis is beyond the scope of this paper, we provide several sufficient conditions for both MNI attainable and unattainable scenarios.

We first present the sufficient conditions under which MNI is attainable. As outlined in Proposition 5, Proposition 6, and Proposition 7, MNI is attainable when the relationships between $X$ and $Y$ are either entirely deterministic or completely independent across each sub-domain. While the deterministic mapping might hold true when $Y$ is the data label, it generally does not hold when $X$ and $Y$ are high-dimensional data modalities. In such cases, a deterministic relationship between the two modalities would imply that one can be fully inferred from the other, leaving no room for modality-specific information in one of the modalities.

**Proposition 5.** *For random variables $X$, $Y$, and representation $Z$ derived from $X$ (i.e., the Markov chain $Z \leftarrow X \leftrightarrow Y$ holds), a sufficient condition for MNI to be attainable is that $X \rightarrow Y$ mapping is deterministic (Fischer, 2020).*

*Proof.* Denote $Y = f(X)$ as the deterministic $X \rightarrow Y$ mapping. If the encoder is powerful enough, it can learn to reproduce the deterministic function $f$, i.e. $Z = f(X) = Y$. Thus $I(X; Z) = I(X; Y) = H(Y) = I(Y; Z)$. $\square$

While Fischer (2020) identifies this as a necessary condition for attainable MNI, there are scenarios where $X \rightarrow Y$ is not deterministic, yet MNI is still attainable. We provide additional sufficient conditions for these more general cases in Proposition 6 and Proposition 7.

**Proposition 6.** *For random variables $X$, $Y$, and representation $Z$ derived from $X$ (i.e., the Markov chain $Z \leftarrow X \leftrightarrow Y$ holds), a sufficient condition for MNI to be attainable is that $Y \rightarrow X$ mapping is deterministic.*

*Proof.* Denote $X = f(Y)$ as the deterministic $Y \to X$ mapping. Then, for any $Z$ with $Z \leftarrow X \leftrightarrow Y$,

$$p(z|y) = \int_x p(z|x)p(x|y)dx = \int_x p(z|x)\mathbf{1}_{x=f(y)}dx = p(z|x = f(y))$$

More formally, $p(z|y) = p(z|X = f(y))$, and

$$p(Y = y, Z = z) = p(Y = y, X = f(y), Z = z) = p(Y = y) \cdot p(z|X = f(y))$$

Thus the conditional entropy can be written as

$$H(Z|Y) = -\int_{Y,Z} p(y,z) \log p(z|y)dydz = -\int_Y p(y) \int_Z p(z|X = f(y)) \log p(z|X = f(y))dydz$$

$$H(Z|X) = -\int_X p(x) \int_Z p(z|x) \log p(z|x)dxdz$$

Thus, $H(Z|Y) = H(Z|X)$, i.e. $I(Y;Z) = I(X;Z)$. By selecting $Z$ such that $I(Z;X) = I(X;Y) = H(X)$, we achieve the MNI point. □

**Proposition 7.** *For random variable $X, Y$ in $\mathcal{X} \times \mathcal{Y}$ with joint distribution $p(X,Y)$, a sufficient condition for the existence of MNI is:*

*In any sub-domain $\mathcal{X}_s \times \mathcal{Y}_s$ of $\mathcal{X} \times \mathcal{Y}$ where $p(X,Y)$ has the full support, i.e. $\forall x, y \in \mathcal{X}_s \times \mathcal{Y}_s, p(x,y) > 0$, $X$ and $Y$ are independent, i.e. $X \perp\!\!\!\perp Y|\{(X,Y) \in \mathcal{X}_s \times \mathcal{Y}_s\}$.*

*Proof.* Construct the learned representation $Z$ as the conditional probability of $Y$ given $X$, i.e. $Z = p(Y|X)$. $Z$ is a deterministic function based o $X$. We will show that such $Z$ satisfies the MNI condition.

First, we show that $I(Z;Y) = I(X;Y)$. Denote $z = p(Y|X = x)$. Since $z$ fully describes the conditional distribution $p(Y|X = x)$, we have $p(Y|X = x) = p(Y|Z = z)$. Thus, $H(Y|X = x) = H(Y|Z = z)$. Therefore,

$$H(Y|X) = \int p(x)H(Y|X = x)dx = \int p(x)H(Y|Z = z)dx$$

$$= \int \left( \int_{\{x:p(Y|X=x)=z\}} p(x)dx \right) H(Y|Z = z)dz = \int p(z)H(Y|Z = z)dz = H(Y|Z)$$

Thus $I(Z;Y) = H(Y) - H(Y|Z) = H(Y) - H(Y|X) = I(X;Y)$.

Second, we show that $I(Z;X) = I(Z;Y)$. Since $I(Z;X) = H(Z) - H(Z|X)$, $I(Z;Y) = H(Z) - H(Z|Y)$, and $H(Z|X) = 0$, $I(Z;X) = I(Z;Y)$ is equivalent to $H(Z|Y) = 0$, i.e. $Z = p(Y|X)$ is determined by the value of $Y$.

The condition is $\forall \mathcal{X}_s \times \mathcal{Y}_s, X \perp\!\!\!\perp Y|\{(X,Y) \in \mathcal{X}_s \times \mathcal{Y}_s\}$, which indicates that

$$\forall \mathcal{X}_s \times \mathcal{Y}_s, \forall x, y \in \mathcal{X}_s \times \mathcal{Y}_s, \frac{p(y|x)}{p(\mathcal{Y}_s|\mathcal{X}_s)} = \frac{p(y)}{p(\mathcal{Y}_s)}$$

For any $y \in \mathcal{Y}$, choose $\mathcal{Y}_s = \{y\}$, and $\mathcal{X}_s = \{x : p(x|y) > 0\}$, we have $p(y|x) = \frac{p(\mathcal{Y}_s|\mathcal{X}_s)}{p(\mathcal{Y}_s)}p(y), \forall x \in \mathcal{X}_s$. Thus, $\forall x_1, x_2$ sampled from $p(X|Y = y)$, we have $p(y|x1) = p(y|x2)$. Therefore, $p(Y = y|X)$ is fully determined given the value of $y$ and is irrelevant to the value of $X$. Then we show that $I(Z;X) = I(Z;Y)$.

It is easy to see that deterministic mapping between $X$ and $Y$ is a special case of this condition.

□

Regarding the sufficient conditions for MNI being unattainable, as outlined in Lemma 6 of Gilad-Bachrach et al. (2003), when variables $X, Y$ have full support, i.e. $p(x,y) > 0, \forall x, y$, MNI is unattainable.

# F DIFFERENTIABILITY OF THE IB CURVE

**Definition 1.** *Let $V$ and $W$ be Banach spaces, $\Omega$ an open set in $V$ and $F$ a function that maps $\Omega$ into $W$. Then the Gateaux derivative of $F$ at $x \in \Omega$ in the direction $h \in V$ is defined as*

$$dF(x;h) = \lim_{\epsilon \to 0} \frac{F(x + \epsilon h) - F(x)}{\epsilon} = \frac{d}{d\epsilon} F(x + \epsilon h)\Big|_{\epsilon=0}$$

*provided that this limit exists for all $h \in V$*

In this section, we use $X$ and $Y$ to represent the two modalities, rather than $X^1$ and $X^2$ in other sections. In our setting, the domain is the Banach space corresponding to the product space representing the set of pairs of joint probability distributions i.e. $\mathcal{Z} \times \mathcal{Y}$. The functional $F = I(Z;Y) : \Omega \to \mathbb{R}$, where $\Omega$ is the product space representing the set of pairs of joint probability distributions for which this mutual information is defined. Following the Markov structure in our graphical model, we have

$$
\begin{aligned}
I(Z;Y) &= \int_{y,z} p(z,y) \log \frac{p(z,y)}{p(z)p(y)} \\
&= \int_{x,y,z} p(z|x,y)p(x|y) \log \frac{\int_x p(z|x,y)p(x|y)}{\int_x p(z|x)p(x)} \\
&= \int_{x,y,z} p(z|x)p(x|y) \log \frac{\int_x p(z|x)p(x|y)}{\int_x p(z|x)p(x)} \\
&= F(f) \text{where } f = p(z|x)
\end{aligned}
$$

**Proposition 8.** *If the MNI point exists, then the Gateaux derivative of $I(Y;Z)$ with respect to $p(z|x)$ (as used in (Pan et al., 2021)) doesn't exist at the MNI point.*

*Proof.* Denote the MNI point between $X$ and $Y$ as $p$ where $I(X;Y) = I(Z;Y) = I(Z;X)$. When the MNI point exists, the IB curve can be represented as (Rodríguez Gálvez et al., 2020; Kolchinsky et al., 2018):

$$
I(Z;Y) = \begin{cases} I(Z;X), & \text{if } I(Z;X) \leq H(Y) \\ H(Y), & \text{if } H(Y) < I(Z;X) \leq H(X) \end{cases}
$$

or equivalently

$$
p(z|y) = \begin{cases} p(z|x), & \text{if } I(Z;X) \leq H(Y) \\ \mathbf{1}_{y=g(z)}, & \text{if } H(Y) < I(Z;X) \leq H(X) \end{cases}
$$

Based on this characterization, we know that there exist a direction $h$ such that on perturbing $p(z|x)$ along this directions, we either have $I(Z;Y) = I(Z;X) < H(Y)$ or $I(Z;Y) = I(X;Y) = H(Y) < I(Z;X)$. Without loss of generality, assume that perturbing by $\epsilon h$, where $\epsilon < 0$ corresponds to the former case and $\epsilon h$, where $\epsilon > 0$ to the latter. Consider the right directional derivative of I(Y:Z) at $p$ in the direction $h$, we have

$$d_+ F(p;h) = \lim_{\epsilon \to 0^+} \frac{F(p + \epsilon h) - F(p)}{\epsilon} = 0$$

Since we have $p(z|y) = p(z|x)$ when $p$ is perturbed in the direction by $\epsilon h$, where $\epsilon < 0$, we can consider $I(Z;Y)$ as an identity function of $I(Z;X)$. In such a case, $F(p + \epsilon h_1) = H(Y) + \epsilon h$. As a consequence, perturbing $p(z|x)$ to get $p'$ is equivalent to the resulting perturbation in $p(z|y)$ also

being $p'$. As a result, we have

$$
\begin{aligned}
d_- F(p; h) &= \lim_{\epsilon \to 0^-} \frac{F(p') - F(p)}{\epsilon} \\
&= \lim_{\epsilon \to 0^-} \frac{F(p(z|y) = p') - F(p)}{\epsilon} \\
&= \frac{d}{d\epsilon} F(p + \epsilon h) \Big|_{\epsilon=0} \\
&= \frac{d}{d\epsilon} \int_{z,y} (p(z|y) + \epsilon h) p(y) \log \frac{p(z|y) + \epsilon h}{\int_y p(z|y) p(y)} \Big|_{\epsilon=0} \\
&= \int_{z,y} h p(y) \log \frac{p(z|y) + \epsilon h}{\int_y p(z|y) p(y)} + (p(z|y) + \epsilon h) p(y) \Big( \frac{h}{p(z|y) + \epsilon h} - \frac{1}{p(z)} \Big) \Big|_{\epsilon=0} \\
&= h \neq 0
\end{aligned}
$$

Clearly, the left and right limits exist but aren't equal, hence proving the non-differentiability.

$\square$

## G  TRACTABLE TRAINING OBJECTIVES

Four terms are involved in DISENTANGLEDSSL, including maximizing the mutual information term $I(Z^1; X^2)$ and minimizing the conditional mutual information term $I(Z^1; X^1|X^2)$ for the shared representations in step 1, as well as maximizing the joint mutual information term $I(Z^1, \hat{Z}_c^{2*}; X^1)$ and minimizing the mutual information term $I(Z^1; \hat{Z}_c^{1*})$ for the modality-specific representations in step 2. We introduce detailed formulations of the tractable training objectives for each of the four terms in this section.

For the inferred shared representations, we model the distributions $\hat{Z}_c^1 \sim p(\cdot|X^1)$ and $\hat{Z}_c^2 \sim p(\cdot|X^2)$ with neural network encoders. Following the common practice in Radford et al. (2021), we use the InfoNCE objective (Oord et al., 2018) as an estimation of the mutual information term $I(Z^1; X^2)$ in $L_c^1$.

$$
L_{\text{InfoNCE}}^c = \mathbb{E}_{z^1, z^{2+}, \{z_i^{2-}\}_{i=1}^N} \left[ -\log \frac{\exp(z^{1\top} z^{2+}/\tau)}{\exp(z^{1\top} z^{2+}/\tau) + \sum_{i=1}^N \exp(z^{1\top} z_i^{2-}/\tau)} \right]
$$

where $\tau$ is the temperature hyperparameter, $z^{2+}$ is the representation of the positive sample corresponding to the joint distribution $p(X^1, X^2)$, and $\{z_i^{2-}\}_{i=1}^N$ are representations of $N$ negative samples from the marginal distribution $p(X^2)$.

We implement the conditional mutual information term $I(Z^1; X^1|X^2)$ in $L_c^1$ using an upper bound developed in Federici et al. (2019), i.e. $I(Z^1; X^1|X^2) \leq D_{\text{KL}}(p(Z^1|X^1)||p(Z^2|X^2))$. While the conditional distributions of representations are modeled as the Gaussian distribution in Federici et al. (2019), we instead use the von Mises-Fisher (vMF) distribution for $p(Z^1|X^1)$ and $p(Z^2|X^2)$ to better align with the InfoNCE objective where the representations lie on the sphere space. Specifically, $\hat{Z}_c^1 \sim \text{vMF}(\mu(X^1), \kappa), \hat{Z}_c^2 \sim \text{vMF}(\mu(X^2), \kappa)$ where $\kappa$ is a hyperparameter controlling for the uncertainty of the representations. Leveraging the formulation of the KL divergence between two vMF distributions, the training objective of $L_c^1$ is to maximize:

$$
L_c^1 = -L_{\text{InfoNCE}}^c + \beta \cdot \mathbb{E}_{x^1, x^2} \left[ \mu(x^1)^\top \mu(x^2) \right]
$$

Note that this objective establish connections with the alignment versus uniformity framework discussed in Wang & Isola (2020), where the conditional information bottleneck constraint corresponds to a higher weight on the alignment term, in contrast to the uniformity term.

The inferred modality-specific representations are encoded as functions $\hat{Z}_s^1 \sim p(\cdot|X^1, \hat{Z}_c^1)$ and $\hat{Z}_s^2 \sim p(\cdot|X^2, \hat{Z}_c^2)$ with deterministic encoders, that takes both data observations and the shared

representations learned in step 1 as input to account for the dependence structure illustrated in Figure 2. The term $I(Z^1, \hat{Z}_c^{2*}; X^1)$ in $L_s^1$ is optimized with the InfoNCE loss, where random augmentations of the data $X^1$ form the two views. Denote the concatenation of $Z^1$ and its corresponding $\hat{Z}_c^{2*}$ as $\tilde{Z}^1$, the InfoNCE objective has the following formula:

$$L_{\text{InfoNCE}}^s = \mathbb{E}_{\tilde{z}^1, \tilde{z}^{1+}, \{\tilde{z}_i^{1-}\}_{i=1}^N} \left[ -\log \frac{\exp(\tilde{z}^{1\top} \tilde{z}^{1+}/\tau)}{\exp(\tilde{z}^{1\top} \tilde{z}^{1+}/\tau) + \sum_{i=1}^N \exp(\tilde{z}^{1\top} \tilde{z}_i^{1-}/\tau)} \right] \quad (10)$$

where $\tilde{z}^{1+}$ is the representation of the positive sample corresponding to the augmented view of $X^1$, and $\{z_i^{1-}\}_{i=1}^N$ are representations of $N$ negative samples from the marginal distribution $p(X^1)$.

For the mutual information term between representations, $I(Z^1; \hat{Z}_c^{1*})$ in $L_s^1$, we implement it as an orthogonal loss to encourage the marginal independence between the shared and modality-specific representations, where the marginal distribution is approximated across a training batch.

$$L_{\text{orthogonal}} = ||[Z_i^1]_{i=1}^{B\top} \cdot [\hat{Z}_{ci}^{1*}]_{i=1}^B||_F \quad (11)$$

where $B$ is the batch size, $[Z_i^1]_{i=1}^B$ and $[\hat{Z}_{ci}^{1*}]_{i=1}^B$ are the concatenations of all the representations in a mini-batch, and $||\cdot||_F$ is the Frobenius norm of the pairwise cosine similarities between each latent dimensions. The training objective of $L_s^1$ in step 2 is to maximize:

$$L_s^1 = -L_{\text{InfoNCE}}^s - \lambda \cdot L_{\text{orthogonal}} \quad (12)$$

## H  EXPERIMENTAL DETAILS AND ADDITIONAL RESULTS

To evaluate the efficacy of our proposed DISENTANGLEDSSL, we conduct a simulation study and two real-world multimodal experiments to address the following key questions:

- How do $\beta$ and $\lambda$ affect DISENTANGLEDSSL's learned representations compared to baselines in a controlled simulation setting with known ground truth?

- Does DISENTANGLEDSSL achieve effective coverage of multimodal information in downstream prediction tasks from MultiBench (Liang et al., 2021), using a combination of shared and modality-specific representations?

- How do DISENTANGLEDSSL's representations perform in high-content drug screening datasets (Subramanian et al., 2017; Cuccarese et al., 2020), assessed by molecule-phenotype retrieval (for the shared) and a disentanglement measurement (for modality-specific)?

Each experiment was conducted on 1 NVIDIA RTX A5000 GPU, each with 24GB of accelerator RAM. All experiments were implemented using the PyTorch deep learning framework.

### H.1  SIMULATION STUDY.

**Experimental details.** We generate synthetic data $X^1$ and $X^2$ based on the graphical model in Figure 2, with the dimensionality of 100 and dataset size being 90,000. To be specific, we sample 50-dimensional true latents $Z_s^1$, $Z_s^2$, and $Z_c$ independently from $\mathcal{N}(\mathbf{0}_{50}, 0.5 \times \mathbf{I}_{50})$. Then we sample the transformation weights $T_1$ and $T_2$ from uniform distribution Uniform$(-1, 1)$, and generate $X^1 = T_1 \cdot [Z_s^1, Z_c]$ and $X^2 = T_2 \cdot [Z_s^2, Z_c]$. We randomly split $80\%$ data into the training set and the rest into the test set. To simulate unattainable MNI, we add Gaussian noise and random dropout during training to ensure the distribution has full support. We use a 3-layer multi-layer perceptron (MLP) with a hidden dimension of 512 as encoders for all methods. For DMVAE, we employ MLPs with the same architecture as decoders. We report the average linear probing accuracy on the test set over 3 random seeds.

We run all the methods on the synthetic data with combinations of different hyperparameter values. For DISENTANGLEDSSL, we use $\beta \in \{0.0, 0.001, 0.01, 0.1, 0.5, 1.0, 5.0, 10.0, 50.0, 100.0, 300.0, 500.0, 1000.0\}$ and $\lambda \in \{0.0, 0.001, 0.01, 0.1, 1.0, 10.0, 100.0\}$.

For JointOpt, hyperparameter $a$ controls the joint mutual information terms $I(\hat{Z}_s^1, \hat{Z}_c^2; X^1)$ and $I(\hat{Z}_s^2, \hat{Z}_c^1; X^2)$, and $\lambda$ adjusts the mutual information term between representations, i.e.

$$\hat{Z}_c^{1*}, \hat{Z}_c^{2*}, \hat{Z}_s^{1*}, \hat{Z}_s^{2*} = \underset{\hat{Z}_c^1, \hat{Z}_c^2, \hat{Z}_s^1, \hat{Z}_s^2}{\arg\max} \quad I(\hat{Z}_c^1; X^2) + I(\hat{Z}_c^2; X^1) + a \cdot (I(\hat{Z}_s^1, \hat{Z}_c^2; X^1) + I(\hat{Z}_s^2, \hat{Z}_c^1; X^2))$$
$$- \lambda \cdot (I(Z_c^1; Z_s^1) + I(Z_c^2; Z_s^2))$$

We use $a \in \{0.01, 0.1, 1.0, 10.0, 100.0, 1000.0\}$ and $\lambda \in \{0.0, 0.001, 0.01, 0.1, 1.0, 10.0, 100.0\}$.

For FOCAL, we tune the hyperparameters $a$ and $\lambda$, defined similarly to JointOpt, where $a$ controls the terms $I(\hat{Z}_s^1; X^1)$ and $I(\hat{Z}_s^2; X^2)$ and $\lambda$ adjusts the orthogonal loss between shared and specific representations. We use the same set of $a$ and $\lambda$ as that in JointOpt.

For DMVAE, we tune $\lambda$ which denotes the weight of the KL divergence term in contrast to the reconstruction loss. We use $\lambda \in \{10^{-7}, 10^{-6}, 10^{-5}, 10^{-4}, 10^{-3}, 10^{-2}\}$.

For SimMMDG, we tune the hyperparameters $a$ and $\lambda$, where $a$ controls the weight of the cross-modal translation loss and $\lambda$ adjusts the distance loss between shared and specific representations. We use the same set of $a$ and $\lambda$ as that in JointOpt.

**Additional results.** We provide a comparison of the performance of the shared representation with baseline methods in Figure 8. Denote $\hat{Z}_c$ as the concatenation of the learned shared representations of $X^1$ and $X^2$, i.e. $\hat{Z}_c = [\hat{Z}_c^1, \hat{Z}_c^2]$. The ideal $\hat{Z}_c$ should maximize the shared information between $X^1$ and $X^2$, shown by high accuracy on $Y_c$, while minimizing unique information of $X^1$ and $X^2$, indicated by low accuracy on $Y_s^1$ and $Y_s^2$. Therefore, a top-left point is preferred in Figure 8.

As shown in Figure 8, DISENTANGLEDSSL consistently outperforms other methods across various hyperparameter settings, showcasing its ability to effectively capture shared information. The only exception occurs in Figure 8a when $\beta$ is set very high and the accuracy on $Y_s^1$ to drop to around 0.50 (equivalent to random guessing, indicating no information of $\hat{Z}_c$ on modality-specific features). In this scenario, DMVAE performs better. This happens because high values of $\beta$ cause the decoder-free contrastive objectives to collapse, with most representations converging to nearly the same point, a commonly-known issue in contrastive self-supervised learning (Oord et al., 2018).

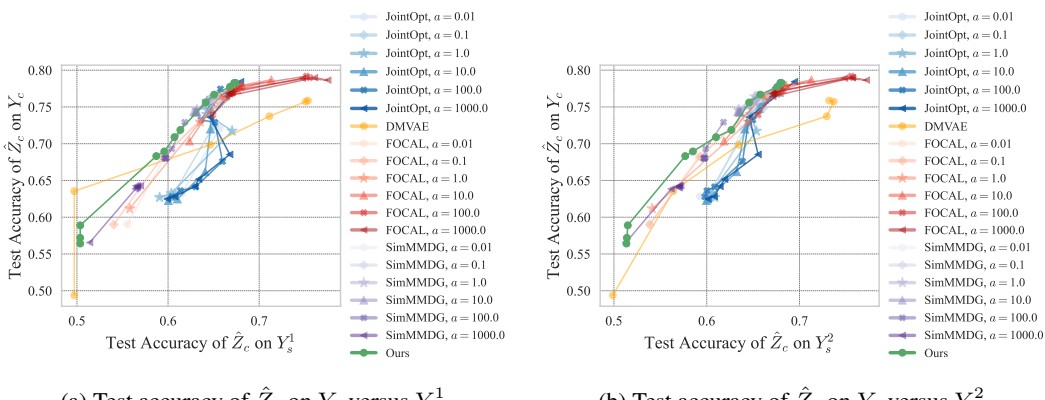

(a) Test accuracy of $\hat{Z}_c$ on $Y_c$ versus $Y_s^1$.    (b) Test accuracy of $\hat{Z}_c$ on $Y_c$ versus $Y_s^2$.

Figure 8: Performance of shared representation $\hat{Z}_c$ for different models.

For the modality-specific representations, as a supplement to Figure 4b, we provide the complete results for both representations $\hat{Z}_s^1$ and $\hat{Z}_s^2$ on a full set of hyperparameters in Figure 9. DISENTANGLEDSSL outperforms all other methods across various hyperparameter settings.

**Analysis for different levels of entanglement.** We further examine the trade-off between expressivity and redundancy for the learned shared representations on synthetic data with varying levels of multimodal entanglement. In this scenario, some dimensions align across modalities with MNI attainable, while others remain entangled with MNI unattainable. Specifically, we split the 50-dimensional true shared latent $Z_c$ into two parts: the first 35 dimensions, denoted as $Z_c^{\text{mix}}$, and the last 15 dimensions, denoted as $Z_c^{\text{pure}}$. We then generate 100-dimensional observations $X^1$ and $X^2$, where

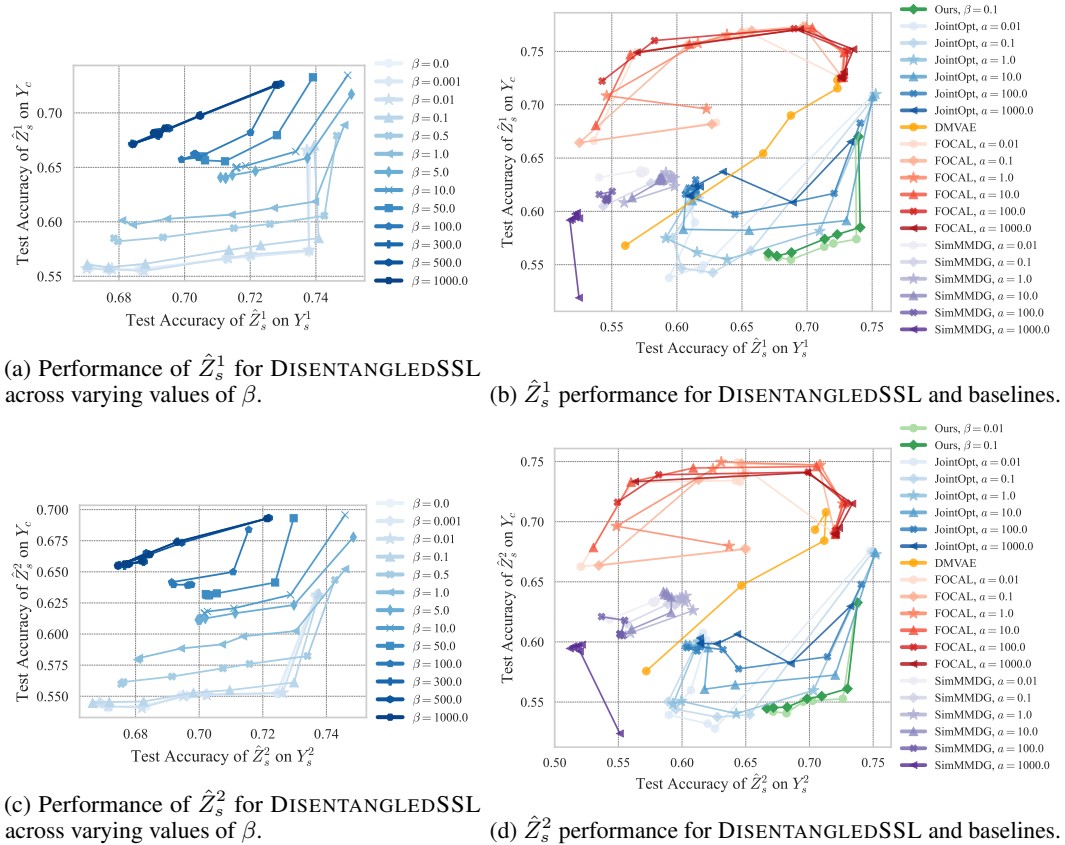

(a) Performance of $\hat{Z}_s^1$ for DISENTANGLEDSSL across varying values of $\beta$.

(b) $\hat{Z}_s^1$ performance for DISENTANGLEDSSL and baselines.

(c) Performance of $\hat{Z}_s^2$ for DISENTANGLEDSSL across varying values of $\beta$.

(d) $\hat{Z}_s^2$ performance for DISENTANGLEDSSL and baselines.

Figure 9: Performance of modality-specific representation $\hat{Z}_s^1$ and $\hat{Z}_s^2$ for different models.

the first 85 dimensions are generated under the same procedure as before, i.e. $X_{\mathrm{mix}}^1 = T_1 \cdot [Z_s^1, Z_c^{\mathrm{mix}}]$ and $X_{\mathrm{mix}}^2 = T_2 \cdot [Z_s^2, Z_c^{\mathrm{mix}}]$ followed by adding Gaussian noise and random dropout, while the last 15 dimensions are directly $Z_c^{\mathrm{pure}}$.

We present the results on the synthetic data with a mixed level of entanglement in Figure 10. Figure 10a shows the test accuracy of $\hat{Z}_c^1$ on $Y_c$, $Y_s^1$, and $Y_s^2$ on the left axis, alongside the MLP weight ratio on $X^1$ and $X^2$ on the right axis, for varying values of $\beta$. To compute the MLP weight ratio, we first extract the diagonal of the inner product of MLP encoder's first layer weight matrix, then calculate the ratio between the average of the last 15 "pure" dimensions and the first 85 "mixed" dimensions. This ratio indicates how much attention the encoder gives to the "pure" versus "mixed" dimensions, with higher values signifying greater focus on the "pure" dimensions. Figure 10b shows the corresponding test accuracy on $Y_c$ versus $Y_s^2$ in line plot.

Using such data with mixed entanglement levels, DISENTANGLEDSSL demonstrates a clear pattern where the learned information plateaus at certain $\beta$ values. As illustrated in Figure 10, the MLP weight ratio initially rises sharply from around 20 to nearly 80, then drops to 1 when $\beta$ becomes very large, indicating the collapse of the learned representations. With a large value of $\beta$ (while before the model collapses, i.e. $\beta \approx 10$), the encoders focus mainly on the "pure" dimensions. This is because a stronger information bottleneck constraint discourages the extraction of shared components from the "mixed" dimensions, which inevitably include modality-specific information due to unattainable MNI, and favors the "pure" shared components with no extra cost.

In addition, we evaluated our method on synthetic data with varying noise levels (i.e. variance of the Gaussian noise and rate of random dropout). As shown in Figure 11, as the noise level decreases, the learned shared representation $\hat{Z}^c$ contains less modality-specific information under the same $\beta$, and the test accuracy on $Y_1$ and $Y_2$ decrease to close to 0.5 even with small $\beta$. Meanwhile, the frontier of

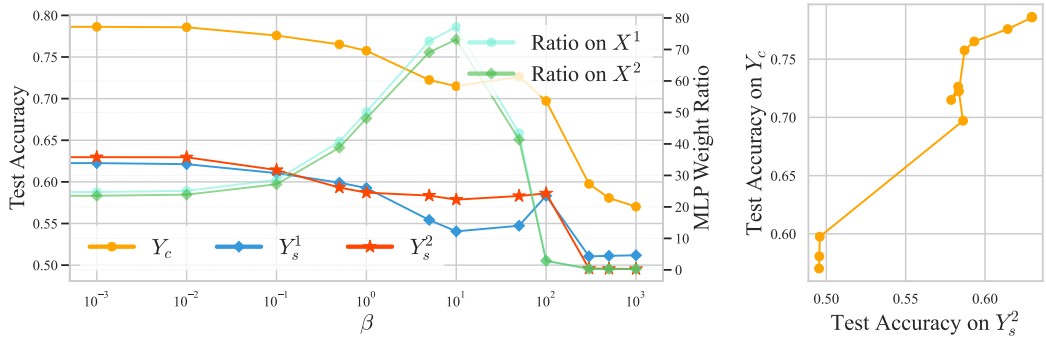

(a) Test accuracy of $\hat{Z}_c^1$ and the MLP weight ratio with varying $\beta$.

(b) Test accuracy on $Y_c$ versus $Y_s^2$.

Figure 10: Performance of $\hat{Z}_c$ on synthetic data with a mixed level of entanglement.

test accuracy on $Y_c$ versus $Y_1$ shrinks towards the top left corner, indicating a decreasing expressivity-redundancy tradeoff and easier disentanglement. These trends align well with our theoretical analysis.

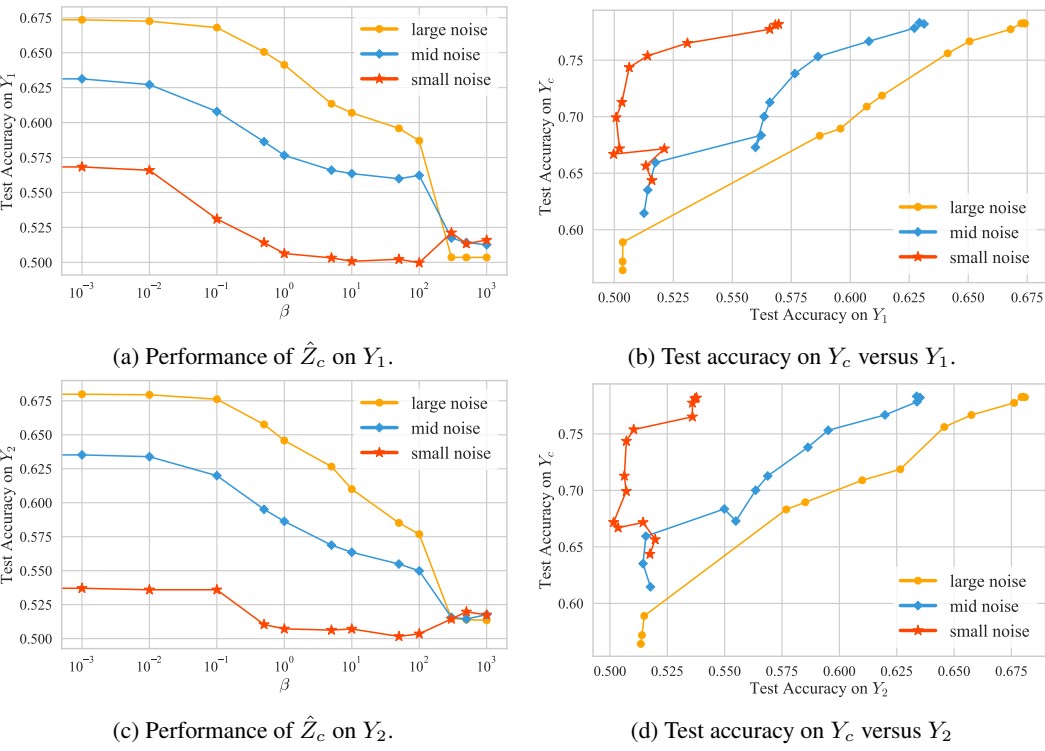

(a) Performance of $\hat{Z}_c$ on $Y_1$.

(b) Test accuracy on $Y_c$ versus $Y_1$.

(c) Performance of $\hat{Z}_c$ on $Y_2$.

(d) Test accuracy on $Y_c$ versus $Y_2$

Figure 11: Performance of shared representation $\hat{Z}_c$ learned with different noise level. As the noise level decreases, $\hat{Z}^c$ contains less modality-specific information, and the frontier of test accuracy on $Y_c$ versus $Y_1$ shrinks toward the top left, reflecting reduced expressivity-redundancy tradeoff.

## H.2 MULTIBENCH.

**Dataset description.** We utilize the real-world multimodal benchmark from MultiBench (Liang et al., 2021). We provide a brief description of each dataset as below.

- **MIMIC** is a large-scale dataset in healthcare, which contains over 40,000 ICU patient records. The two modalities are the time series patient records measured by hours and tabular static data (age, gender, etc.). The binary task label is on whether the patient fits any ICD-9 code in group 7.

- **MOSEI** is a sentence-level multimodal sentiment and emotion benchmark with 23,000 monologue videos. The two modalities are the vision and text modalities, and the label is on whether the sentiment is positive or negative corresponding to each video.

- **MOSI** is a similar multimodal sentiment analysis dataset based on 2,199 YouTube video clips, with vision and text modalities and the sentiment label.

- **UR-FUNNY** is a humor detection dataset in human speech, which consists of samples from TED talk videos, with vision and text modalities and a binary label on whether there is humor or not.

- **MUSTARD** is a corpus of 690 videos from popular TV shows for sarcasm detection. The vision and text modalities are utilized to classify a binary label of sarcasm.

**Experimental details.** We follow the same dataset splitting, and utilize the same encoder architecture and pre-extracted features as Liang et al. (2024). All models use representations (or concatenation of them, for FactorCL-proj, FOCAL, JointOpt and DISENTANGLEDSSL) with the same dimensionality. We set the latent dimension to 300 across all datasets, except for MUSTARD, where it is set to 100 due to the smaller dataset size, and for MIMIC, where we use 360 to align with the output dimension of the GRU encoders. For FactorCL, we use their default hyperparameter settings. For other methods, hyperparameters are tuned based on validation set performance. For DISENTANGLEDSSL, we use $\beta = 1.0$ and $\lambda = 10^{-3}$ for all datasets, except for MOSI where $\beta = 0.01$. For FOCAL and JointOpt, we use $a = 1$ and $\lambda = 10^{-3}$ across all datasets. We report the mean and standard deviation of the linear probing accuracy on prediction labels from the test set over 3 random seeds.

Table 4: Prediction accuracy (%) of the representations learned by different methods on MultiBench datasets and standard deviations over 3 random seeds.

| Dataset | MIMIC | MOSEI | MOSI | UR-FUNNY | MUSTARD |
|---|---|---|---|---|---|
| CLIP | 64.97 (0.60) | 76.87 (0.45) | 64.24 (0.88) | 62.73 (0.92) | 56.04 (4.19) |
| FactorCL-emb | 65.25 (0.45) | 71.80 (0.64) | 62.97 (0.81) | 63.29 (2.07) | 56.76 (4.66) |
| FactorCL-proj | 59.43 (1.70) | 74.61 (1.65) | 56.02 (1.26) | 61.25 (0.47) | 55.80 (2.18) |
| FOCAL (shared) | 62.69 (1.46) | 75.93 (0.23) | 62.10 (0.44) | 62.38 (0.58) | 54.83 (3.02) |
| FOCAL (specific) | 62.13 (1.49) | 75.41 (0.54) | 61.61 (1.27) | 63.17 (0.96) | 58.21 (2.21) |
| FOCAL (both) | 64.42 (0.34) | 76.77 (0.51) | 63.65 (1.09) | 62.98 (1.52) | 54.35 (0.00) |
| JointOpt (shared) | 63.01 (0.59) | 76.69 (0.28) | 65.02 (1.96) | 62.51 (1.02) | 54.83 (4.82) |
| JointOpt (specific) | 65.81 (0.49) | 74.40 (0.94) | 53.89 (0.80) | 62.13 (0.69) | 57.73 (4.12) |
| JointOpt (both) | 66.11 (0.64) | 76.71 (0.14) | 64.24 (1.75) | 63.58 (1.45) | 56.52 (2.61) |
| DISENTANGLEDSSL (shared) | 63.16 (0.48) | 76.94 (0.22) | **65.16** (0.81) | 64.14 (1.53) | 54.11 (1.51) |
| DISENTANGLEDSSL (specific) | 65.73 (0.09) | 75.99 (0.60) | 51.70 (0.72) | 60.27 (1.28) | **61.60** (2.61) |
| DISENTANGLEDSSL (both) | **66.44** (0.31) | **77.45** (0.06) | 65.11 (0.80) | **64.24** (1.54) | 56.52 (2.18) |

**Additional results.** As a complement to Table 1, we present results of FOCAL and JointOpt for the learned shared and modality-specific representations, both combined (i.e. "both") and separately in Table 4. Although they exhibit a similar trend as DISENTANGLEDSSL, where shared representations are more important for MOSI and specific representations are crucial for MUSTARD, the distinction is less pronounced. Moreover, the overall performance is inferior to that of DISENTANGLEDSSL, indicating that DISENTANGLEDSSL achieves a better balance between coverage and disentanglement in the learned representations.

**Ablation study.** We conduct an ablation study of DISENTANGLEDSSL on the MultiBench datasets to examine the impact of the hyperparameters $\beta$ and $\lambda$ in Table 5. As $\beta$ increases, the shared representations preserve only the most closely shared information and discard other features. Therefore, on datasets where the loosely shared information contributes to the label (eg., UR-FUNNY, MOSI, MIMIC), the prediction accuracy decreases with increasing $\beta$. For the MUSTARD dataset where there exist contradictions between language and video (Liang et al., 2024), the accuracy of the shared representations increases with very high $\beta$ since the contradicted parts are discarded and only clear signals remain. Similarly, as $\lambda$ increases, the independence constraint between the shared and specific representations is emphasized more. Thus, the modality-specific representations contains less information, potentially related to the label, making their prediction accuracy decrease.

Table 5: Prediction accuracy (%) of the representations learned by DISENTANGLEDSSL with different $\beta$ and $\lambda$ and standard deviations over 3 random seeds.

| Dataset | $\beta$ | $\lambda$ | Shared | Specific | Both |
|---|---|---|---|---|---|
| UR-FUNNY | 0.01 | 0.001 | 64.08 (1.48) | 62.04 (0.66) | 64.43 (1.14) |
| | 1.0 | 0.0 | 64.14 (1.53) | 61.81 (0.08) | 64.24 (0.96) |
| | 1.0 | 0.001 | 64.14 (1.53) | 60.27 (1.28) | 64.24 (1.54) |
| | 1.0 | 0.1 | 64.14 (1.53) | 60.46 (0.32) | 64.46 (0.74) |
| | 100.0 | 0.001 | 62.70 (0.70) | 58.76 (0.35) | 61.72 (0.83) |
| MOSI | 0.0001 | 0.001 | 65.16 (0.60) | 51.70 (0.77) | 64.97 (0.34) |
| | 0.01 | 0.0 | 65.16 (0.81) | 51.36 (1.37) | 64.97 (0.50) |
| | 0.01 | 0.001 | 65.16 (0.81) | 51.70 (0.72) | 65.11 (0.80) |
| | 0.01 | 0.1 | 65.16 (0.81) | 50.53 (1.31) | 64.72 (0.83) |
| | 1.0 | 0.001 | 63.31 (0.70) | 50.87 (1.15) | 63.95 (0.77) |
| MIMIC | 0.01 | 0.001 | 62.73 (0.26) | 66.05 (0.39) | 66.25 (0.36) |
| | 1.0 | 0.0 | 63.16 (0.48) | 65.74 (0.20) | 66.26 (0.35) |
| | 1.0 | 0.001 | 63.16 (0.48) | 65.73 (0.09) | 66.44 (0.31) |
| | 1.0 | 0.1 | 63.16 (0.48) | 65.37 (0.48) | 66.38 (0.19) |
| | 100.0 | 0.001 | 61.94 (0.29) | 62.82 (0.85) | 62.70 (0.63) |
| MOSEI | 0.01 | 0.001 | 76.68 (0.21) | 75.71 (0.43) | 77.00 (0.14) |
| | 1.0 | 0.0 | 76.94 (0.22) | 75.97 (0.52) | 77.54 (0.19) |
| | 1.0 | 0.001 | 76.94 (0.22) | 75.99 (0.60) | 77.45 (0.06) |
| | 1.0 | 0.1 | 76.94 (0.22) | 75.83 (0.39) | 77.56 (0.11) |
| | 100.0 | 0.001 | 77.20 (0.16) | 75.35 (0.72) | 77.26 (0.27) |
| MUSTARD | 0.01 | 0.001 | 56.28 (1.90) | 58.70 (1.57) | 55.80 (2.05) |
| | 1.0 | 0.0 | 54.11 (1.51) | 60.14 (0.59) | 56.28 (4.16) |
| | 1.0 | 0.001 | 54.11 (1.51) | 61.60 (2.61) | 56.52 (2.18) |
| | 1.0 | 0.1 | 54.11 (1.51) | 61.35 (2.08) | 57.49 (1.49) |
| | 100.0 | 0.001 | 57.25 (3.13) | 61.84 (4.78) | 57.97 (2.05) |

### H.3 HIGH-CONTENT DRUG SCREENING

**Experimental details.** Following the setup in Wang et al. (2023a), we utilize Mol2vec (Jaeger et al., 2018) to featurize the molecular structures into 300-dimensional feature vectors. For both molecular structures and phenotypes, we employ 3-layer MLP encoders with a hidden dimension of 2560. The dimensionality for shared and modality-specific representations is set to 32 across all methods and datasets. We tune $\beta$ based on validation set performance, with $\beta = 5.0$ for RXRX19a and $\beta = 1.0$ for LINCS, and set $\lambda = 0.01$ for both datasets. For FOCAL and JointOpt, we set $a = 1.0$ and $\lambda = 0.01$. For DMVAE, we set the coefficient of the KL divergence term as $10^{-5}$. In addition, as in Lee & Pavlovic (2021), we introduce an InfoNCE loss for the shared representations to DMVAE in the high-content drug screening experiments to enhance its retrieving accuracy. We report the mean and standard deviation of all results on the test set over 3 random seeds.

**Counterfactual generation.** To further evaluate the learned modality-specific representations, we use different combinations of the shared and modality-specific representations to generate counterfactual samples, i.e. predicting phenotype of a drug on a different cell with its molecule shared latent but the phenotype specific latent of a different cell perturbed by other drugs. Since the factual observations of counterfactual generation are unavailable, we measure the performance distributional-wise, introducing the difference in Frechet distance (Diff-FD) as a metric, i.e. Diff-FD-c measuring the gain of using the correct batch of molecules, thus the information of the shared latent; Diff-FD-s measures the gain of using the correct batch of cells, thus the information of the specific latent.

To be specific, we train decoders with the factual combination of the shared latents from molecular structures and the modality-specific latents from phenotypes on the training set. We use "val recon" to denote the samples generated using the representations attained from validation set molecules and validation set phenotypes as input. Similarly, "test recon" refers to the samples generated using test set molecules paired with test set phenotypes, while "counterfactual" represents those generated using test set molecules paired with validation set phenotypes. Formally, Diff-FD-c is defined as FD(val recon; test recon) - FD(counterfactual; test recon), and Diff-FD-s as FD(val recon; test recon) - FD(counterfactual; val recon). Note that a non-informative specific latent can lead to very high Diff-FD-c, and an overly informative modality-specific latent can lead to very high Diff-FD-s, thus we take both metrics into consideration together.

As shown in Table 6, DISENTANGLEDSSL outperforms JointOpt in both metrics and surpasses FOCAL in Diff-FD-s. While FOCAL shows a high Diff-FD-s, it learns overly informative modality-specific latents, which contains significant shared information, as highlighted in the simulation study results (i.e. the modality-specific representations of FOCAL have high accuracy on $Y_c$ in Figure 9), leading to a low value of Diff-FD-c. In contrast, DISENTANGLEDSSL demonstrate better effectiveness in capturing both shared and modality-specific information, while maximizing the separation between them, as indicated by its high values in both metrics.

Table 6: Results on counterfactual generation.

| Dataset | RXRX19a | | LINCS | |
|---|---|---|---|---|
| Metric | Diff-FD-c | Diff-FD-s | Diff-FD-c | Diff-FD-s |
| FOCAL | 15.30(0.85) | **22.89**(0.28) | 0.246(0.039) | **0.765**(0.027) |
| JointOpt | 20.34(0.47) | 9.40(0.39) | 0.248(0.029) | 0.333(0.050) |
| DISENTANGLEDSSL | **20.92**(0.71) | 11.36(0.36) | **0.277**(0.031) | 0.401(0.085) |

## I   ALGORITHM PSEUDOCODE

---

**Algorithm 1** PyTorch-based pseudocode for DISENTANGLEDSSL

---

1: **Notations:** $f_c^1$, $f_c^2$ represents the backbone encoder network for the shared representations. $f_s^1$, $f_s^2$ represents the backbone encoder network for the modality-specific representations. $\beta$ and $\lambda$ are the Lagrangian weights for the shared and modality-specific objectives respectively. vMF represents von Mises-Fisher sampling.

2:   /* Step 1: Learn shared representations */

3: **for** minibatch $(x^1, x^2)$ in dataloader **do**

4:     $\hat{z}_c^1 \sim \text{vMF}(f_c^1(x^1), \kappa)$, $\hat{z}_c^2 \sim \text{vMF}(f_c^2(x^2), \kappa)$

5:     /* Compute InfoNCE loss for shared representations */

6:     $\mathcal{L}_{\text{InfoNCE}}^c = \text{infonce\_loss}(\hat{z}_c^1, \hat{z}_c^2)$

7:     /* Step 1: Objective for shared representations */

8:     $\mathcal{L}_c = \mathcal{L}_{\text{InfoNCE}}^c - \beta \cdot \mathbb{E}[f_c^1(x^1)^\top f_c^2(x^2)]$

9:     $\mathcal{L}_c$.backward()

10:     step_1_optimizer.step()

11:   /* Step 2: Learn modality-specific representations */

12: **for** minibatch $(x^1, x^2)$ in dataloader **do**

13:     Generate augmented views of the data $(x_a^1, x_a^2)$ and $(x_b^1, x_b^2)$

14:     /* Calculate the learned shared representations */

15:     $\hat{z}_{c,a}^1 \leftarrow f_c^1(x_a^1)$, $\hat{z}_{c,a}^2 \leftarrow f_c^2(x_a^2)$, $\hat{z}_{c,b}^1 \leftarrow f_c^1(x_b^1)$, $\hat{z}_{c,b}^2 \leftarrow f_c^2(x_b^2)$

16:     /* Encode modality-specific representations conditioned on shared components */

17:     $\hat{z}_{s,a}^1 \leftarrow f_s^1(x_a^1, \hat{z}_{c,a}^1)$, $\hat{z}_{s,a}^2 \leftarrow f_s^2(x_a^2, \hat{z}_{c,a}^2)$, $\hat{z}_{s,b}^1 \leftarrow f_s^1(x_b^1, \hat{z}_{c,b}^1)$, $\hat{z}_{s,b}^2 \leftarrow f_s^2(x_b^2, \hat{z}_{c,b}^2)$

18:     /* Compute modality-specific losses */

19:     $\mathcal{L}_{\text{InfoNCE}}^s = \text{infonce\_loss}(\text{concat}[\hat{z}_{s,a}^1, \hat{z}_{c,a}^2], \text{concat}[\hat{z}_{s,b}^1, \hat{z}_{c,b}^2]) + \text{infonce\_loss}(\text{concat}[\hat{z}_{s,a}^2, \hat{z}_{c,a}^1], \text{concat}[\hat{z}_{s,b}^2, \hat{z}_{c,b}^1])$

20:     $\mathcal{L}_{\text{orth}} = \frac{1}{2} \sum_{x \in \{a,b\}} \sum_{i \in \{1,2\}} \text{orthogonal\_loss}(\hat{z}_{s,x}^i, \hat{z}_{c,x}^i)$

21:     $\mathcal{L}_s = \mathcal{L}_{\text{InfoNCE}}^s + \lambda \cdot \mathcal{L}_{\text{orth}}$

22:     $\mathcal{L}_s$.backward()

23:     step_2_optimizer.step()

---

## J   NOTATIONS

Table 7: Notations and mathematical definitions

| Variable(s) | Definition |
| --- | --- |
| *Variables of Data Generation* | |
| $\{X^1, X^2\}$ | Paired observation or data from the two modalities (such as image and text pairs) |
| $Z_c$ | Shared latent representation capturing common information |
| $Z_s^i$ | Modality-specific latent representations for observations $X_i$ |
| *Inferred representations* | |
| $\hat{Z}_c^i$ | Shared latent representation inferred only from $X_i$ i.e. $\hat{Z}_c^i \sim p(\cdot|X^i)$ |
| $\hat{Z}_s^i$ | Modality-specific latent representations inferred jointly from observations $X_i$ and the inferred shared representations $\hat{Z}_c^i$ i.e. $\hat{Z}_s^i \sim p(\cdot|X^i, \hat{Z}_c^i)$ |
| $\hat{Z}_c^{i*}$ | Optimal shared latent representation inferred only from $X_i$ |
| $\hat{Z}_s^{i*}$ | Optimal modality-specific latent representation inferred jointly from observations $X_i$ and the optimal shared representations $\hat{Z}_c^{i*}$ |
| *Optimization Parameters* | |
| $\delta_c$ | Small tolerance to account for non-attainability of MNI for the shared representation |
| $\delta_s$ | Small tolerance for coverage-disentanglement trade-off for modality-specific latent? |
| $\beta$ | Lagrangian coefficient controlling trade-offs in optimization for the shared representation |
| $\lambda$ | Lagrangian coefficient controlling trade-offs in optimization for the modality-specific representation |
| $L_c^i$ | Objective to be maximized for learning the shared representation using $X_i$ |
| $L_s^i$ | Objective to be maximized for the learning the modality-specific representation for modality $i$ |
| *Standard Notations and Definitions* | |
| MNI | Minimum Necessary Information |
| $I(\cdot;\cdot)$ | Mutual information between variables |
| $\mathcal{N}(\mu, \Sigma)$ | Multivariate normal distribution |
| vMF$(\mu, \kappa)$ | Von Mises-Fisher distribution used for modeling representations |

