# OpenReview forum: "An Information Criterion for Controlled Disentanglement of Multimodal Data"
_ICLR.cc/2025/Conference — ICLR 2025 Poster_

### Official Review · Reviewer_bzKx · 2024-11-02

**Soundness:** 4
**Presentation:** 3
**Contribution:** 4
**Rating:** 6
**Confidence:** 3

**Summary:**

This paper addresses the disentanglement of modality-shared and modality-specific information in multi-modal self-supervised learning and proposes DisentangledSSL. From an information-theoretic perspective, the modality-shared information is optimized using a conditional entropy bottleneck (CEB). Correspondingly, the authors formulate an optimization problem to isolate modality-specific information. Theoretical analysis of the optimality of each disentangled representation, particularly when Minimum Necessary Information is unattainable, along with experimental results, demonstrates the superiority of DisentangledSSL.

**Strengths:**

1.The idea of applying conditional entropy bottleneck (CEB) in multi-modal self-supervised learning is novel.

2.Comprehensive theoretical analysis is conducted to prove the optimality of both modality-specific and modality-shared information.

3.DisentangledSSL demonstrates superior performance across tasks, including vision-language prediction and molecule-phenotype retrieval.

**Weaknesses:**

1.How does the optimization objective for the modality-shared representation change from Eq. (1) to Eq. (3)? Why is $I(X^1;X^2)$ omitted?

2.Related studies, such as CoCoNet [1] and SimMMDG [2], which also address the separation of modality-shared and modality-specific information, are recommended for inclusion and comparison. Note that SimMMDG [2] can be easily adapted to a self-supervised setting by substituting supervised contrastive learning with the original self-supervised counterpart.

3.Ablation studies on the impact of $\beta$ and $\lambda$ in the MultiBench datasets should be provided to demonstrate the importance of separating modality-specific and modality-shared information in real-world applications.

4.Some typos.
1)$Z^2$ should be $X^2$ in Eq.(6) and in the equation below Eq.(6).
2)$I(X^1;Z^1)$ should be $I(X^1;X^2)$ in Figure 3.

[1] Li J, Qiang W, Zheng C, et al. Modeling multiple views via implicitly preserving global consistency and local complementarity. TKDE, 2022.
[2] Dong H, Nejjar I, Sun H, et al. SimMMDG: A simple and effective framework for multi-modal domain generalization. NIPS, 2023.

**Questions:**

Are there any alternative approaches for implementing the $I(Z^1;\hat{Z}_{c}^{1*})$ in $L_s^1$? If so, how does their performance compare to that of the orthogonal loss implementation?

---

> ### Author Response · Authors · 2024-11-19
> **Response to Reviewer bzKx**
>
> We would like to thank the reviewer for their diligent and insightful review and their encouraging words. We have incorporated the reviewer’s comments in the updated paper, with the changes in the manuscript highlighted in magenta. We share our thoughts on the questions asked below.
>
>
> > How does the optimization objective for the modality-shared representation change from Eq. (1) to Eq. (3)? Why is $I(X^1;X^2)$ omitted?
>
> The Lagrangian form of the constraint optimization problem in Eq. (1) has the formula
> $$\tilde{L} = I(Z^1;X^1|X^2) + \tilde{\beta} \cdot (I(X^1;X^2) - I(Z^1;X^2))$$
> Since we are optimizing for the representation $Z^1$, with given $X^1$ and $X^2$, the term $I(X^1;X^2)$ is a fixed constant irrelevant to $Z^1$ and can be omitted from the objective, i.e.
> $$\arg \min_{Z^1} \tilde{L} = \arg \min_{Z^1} I(Z^1;X^1|X^2) - \tilde{\beta} \cdot I(Z^1;X^2)$$
> This is equivalent to the objective in Eq. (3), i.e.
> $$\arg \min_{Z^1} \tilde{L} = \arg \max_{Z^1} L_c^1 = \arg \max_{Z^1} I(Z^1;X^2) - \beta \cdot I(Z^1;X^1|X^2)$$
>
> Additionally, the properties of the optimal representations learned by Eq. (3) are more rigorously justified in Proposition 1 and Proposition 2.
>
> > Related studies, such as CoCoNet and SimMMDG, are recommended for inclusion and comparison.
>
> We have included CoCoNet and SimMMDG in the updated Section 4 with discussions. Also, we experiment with SimMMDG in the simulation study, with results shown in the updated Figure 4b, Figure 8, and Figure 9. DisentangledSSL outperforms SimMMDG in this setting.
>
> Besides, although the methods in CoCoNet and SimMMDG are related to DisentangledSSL, their specific problem settings are quite different from ours. CoCoNet focuses on learning image representations via modeling multiple views of one image (i.e. L, RGB, ab). SimMMDG focuses on a domain generalization setting with supervised labels. In contrast, we focus on the self-supervised setting with general multimodal data.
>
> > Ablation studies on the impact of $\beta$ and $\lambda$ in the MultiBench datasets should be provided to demonstrate the importance of separating modality-specific and modality-shared information in real-world applications.
>
> Thank you for your valuable suggestion. We have added an ablation study of DisentangledSSL on the MultiBench datasets to examine the impact of the hyperparameters $\beta$ and $\lambda$ in Table 5. Further, the effectiveness of separating modality-specific and modality-shared information is already reflected in the superior performance of DisentangledSSL compared to other baselines.
>
> > Some typos. 1) $Z^2$ should be in $X^2$ Eq.(6) and in the equation below Eq.(6). 2) $I(X^1;Z^1)$ should be $I(X^1; X^2)$ in Figure 3.
>
> Thanks a lot for pointing out the typo in Equation (6). We have corrected it in the revised manuscript. However, $I(X^1; Z^1)$ is correct in Figure 3. It indicates the values in the horizontal axis and is measured for different $Z^1$.
>
> > Are there any alternative approaches for implementing the $I(Z^1; \hat{Z}_c^{1*})$ in $L_s^1$? If so, how does their performance compare to that of the orthogonal loss implementation?
>
> That's a great point! Yes, there are alternative approaches for implementing the mutual information term $I(Z^1; \hat{Z}_c^{1*})$ in $L_s^1$. One such approach is adversarial training (eg., [1,2]), whereby an encoder and discriminator play a minimax game. However, adversarial training can be unstable and difficult to tune, which makes it less practical for some applications. Other approaches include applying measures like the Hilbert-Schmidt Independence Criterion (HSIC) [3] or Distance Correlation as regularizers to encourage statistical independence.
>
> In contrast, we use an orthogonal loss, which is a simple and effective method to minimize the mutual information between $Z^1$ and $\hat{Z}_c^{1*}$. That said, DisentangledSSL can accommodate different implementations of each term in the loss function, allowing for easy integration of future advances in mutual information optimization.
>
> [1] Disentangled Information Bottleneck. AAAI 2021.
>
> [2] Learning disentangled representations via mutual information estimation. ECCV 2020.
>
> [3] A Kernel Statistical Test of Independence. NeurIPS 2007.

---

> ### Comment · Reviewer_bzKx · 2024-11-26
> **response**
>
> Thanks for the responses. My concerns are sufficiently addressed, and I'd like to maintain my positive rating.

---

> ### Author Response · Authors · 2024-11-28
> **Thanks to Reviewer bzKx**
>
> Thank you for your response and insightful review!

---

### Official Review · Reviewer_DHvL · 2024-11-03

**Soundness:** 2
**Presentation:** 2
**Contribution:** 2
**Rating:** 6
**Confidence:** 4

**Summary:**

The authors believe that "shared information between modalities is precisely what is relevant for downstream tasks" and "the modality gap ... results in misalignment between modalities, restricting the application of these methods in numerous real-world multimodal scenarios." Hence they motivate "the need for a disentangled representation space that captures both shared and modality-specific information". To address this need, they propose DISENTANGLEDSSL, an information-theoretic framework designed to learn disentangled shared and modality-specific representations of multimodal data. The authors have conducted a theoretical analysis to evaluate the quality of disentanglement, which can be applied even in cases where Minimum Necessary Information (MNI) is unattainable. The efficacy of DISENTANGLEDSSL is demonstrated across a range of synthetic and real-world multimodal datasets and tasks, including prediction tasks for vision-language data and molecule-phenotype retrieval tasks for biological data.

**Strengths:**

It makes sense to learn disentangled shared and modality-specific representations of multimodal data.

The theoretical analysis to evaluate the quality of disentanglement, which can be applied even in cases where Minimum Necessary Information (MNI) is unattainable, is provided.

The efficacy of DISENTANGLEDSSL is demonstrated across a range of synthetic and real-world multimodal datasets and tasks, including prediction tasks for vision-language data and molecule-phenotype retrieval tasks for biological data.

**Weaknesses:**

Some of the key assertions motivating this work are inaccurate, and the proposed graphical model is flawed, which together raise questions about the validity of the resulting framework.

For instance, the statement "shared information between modalities is precisely what is relevant for downstream tasks" is not universally accurate. What is relevant depends heavily on the specific downstream tasks. Furthermore, "shared information" is a vague term, especially when applied across modalities, making it problematic in this context.

The assertion that "the modality gap ... results in misalignment between modalities, restricting the application of these methods in numerous real-world multimodal scenarios" is also oversimplified. The modality gap is not the sole or primary cause of misalignment; other factors such as alignment methods, distributional shifts, domain shifts, sampling biases, and more can contribute significantly.

I suggest the authors to refine their motivation to more accurately reflect the complexities of multimodal relationships, while still maintaining the core idea of disentangling shared and modality-specific information.

The graphical model in Figure 2, intended to represent the generative process (as claimed in line 128), has two major issues. First, the authors have conflated the generative and inference processes. Only the pathway from Z to X represents the generative model, while the path from X to \hat{Z} corresponds to inference, which should not be included in the data-generating graphical model. This issue is straightforward to resolve by removing the inference process from the model, which would not affect the algorithm or results, or to create separate diagrams, one for the generative process and the other for inference to clarify.

The second issue is more severe: the assumption of a shared latent variable Z_c existing between two modalities may not hold. This assumption lacks foundation, as it is more likely that the relationship takes the form of two variables Z_c^1 and Z_c^2, with potential connections between them:  (1)  Z_c^1 -> Z_c^2, (2)  Z_c^1 <- Z_c^2, or (3) or Z_c^1 <-C-> Z_c^2, depending on the modalities involved. This discrepancy significantly undermines the justification for the proposed algorithm.  I encourage the authors to discuss the implications of using separate but potentially connected latent variables (Z_c^1 and Z_c^2) instead of a single shared Z_c, and how might this change affect the proposed algorithm and results. This could lead to a more nuanced and realistic model of multimodal relationships.

**Questions:**

Please respond to the weaknesses listed above.

**Details Of Ethics Concerns:**

n.a.

---

> ### Author Response · Authors · 2024-11-19
> **Response to Reviewer DHvL**
>
> We are grateful to the reviewer for the time they put in to review our work. We are glad to see that they recognize several strengths in our work, including theoretical analysis based on the unattainability of MNI and comprehensive empirical evaluation across a range of synthetic and real-world multimodal datasets. Below, we share our thoughts on the questions asked and have updated the manuscript with changes highlighted in magenta.
>
>
> > The statement "shared information between modalities is precisely what is relevant for downstream tasks" is not universally accurate. Furthermore, "shared information" is a vague term, especially when applied across modalities, making it problematic in this context.
>
>
> It appears there may be some misunderstanding. This statement is used in our paper to point out the unrealistic assumption of “multi-view redundancy” exploited by previous works (e.g., [1,2,3]) that only maximizes the mutual information between modalities. “Shared information” in these contexts indicates the information that is common and redundant to all modalities. This assumption is a limitation of these prior methods, and our work does not adopt it. Instead, we consider the general case where downstream tasks are related to both shared and modality-specific information.
>
> [1] An information theoretic framework for multi-view learning. COLT 2008.
>
> [2] Contrastive learning, multi-view redundancy, and linear models. Algorithmic Learning Theory 2021.
>
> [3] Self-supervised learning from a multi-view perspective. ICLR 2020.
>
>
> > The modality gap is not the sole or primary cause of misalignment; other factors, such as alignment methods, distributional shifts, domain shifts, sampling biases, and more, can contribute significantly.
>
> We agree with the reviewer that the modality gap is not the sole cause of misalignment between modalities; many factors including alignment methods, distributional shifts, domain shifts, and so on, also contribute. However, the modality gap is one of these key factors and serves as the primary motivation for our method that learns both shared and modality-specific representations. While we acknowledge that other factors can cause misalignment, addressing them (all) is not the main focus of our study here.
>
> > I suggest the authors refine their motivation to more accurately reflect the complexities of multimodal relationships, while still maintaining the core idea of disentangling shared and modality-specific information.
>
> In response to the reviewer’s suggestion, we have expanded our introduction to more thoroughly discuss the complexities of multimodal relationships. Please find the revisions indicated in magenta color in the revised manuscript.

---

> ### Author Response · Authors · 2024-11-19
> **Response to Reviewer DHvL (continued)**
>
> >  The authors have conflated the generative and inference processes in the graphical model in Figure 2.
>
> Thank the reviewer for this suggestion. We have updated the graphical model with a separate generative process and inference process in the revised paper. Nevertheless, it has become a common practice in previous papers to combine generation and inference parts in the same graphical model when there’s little chance for confusion. The graphs are used as illustrations, not intended as formal graphical models. For example, you would see this in [4] in the context of multimodal representation learning, and [5] in the context of diffusion models.
>
> [4] Private-Shared Disentangled Multimodal VAE for Learning of Latent Representations. CVPR 2021.
>
> [5] Denoising Diffusion Probabilistic Models. NeurIPS 2020.
>
>
> > The assumption of a shared latent variable $Z_c$ existing between two modalities may not hold.
>
> In this paper, we focus on modeling the statistical relationships between the domains rather than causal relationships. Therefore, the three graphical relations between variables proposed by the reviewer are statistically (marginally) equivalent, and are in our case represented by a single variable $Z_c$ (variability specific to a domain would be delegated to other variables). This statistical setup is not unique to our paper but commonly used in many past works that also learn disentangled representations of multimodal data, for example, [6,7,8,9,10,11]. These involve data generation processes with a single shared latent variable and modality-specific latent variables corresponding to each modality.
>
> We emphasize that the focus and the main contribution of our paper is not to study a causal graphical model but rather on how to extract and infer the representations from the data. In particular, we address the challenging real-world setting where Minimum Necessary Information (MNI) is not attainable. This case is underexplored in prior works. We provide detailed and rigorous analysis in Sections 2.2 and 2.3 and demonstrate the efficacy of our proposed method through comprehensive experiments in Section 3.
>
> [6] Private-Shared Disentangled Multimodal VAE for Learning of Latent Representations. CVPR 2021.
>
> [7] Self-supervised Disentanglement of Modality-specific and Shared Factors Improves Multimodal Generative Models. Pattern Recognition 2020.
>
> [8] Variational Interaction Information Maximization for Cross-domain Disentanglement. NeurIPS 2020.
>
> [9] FOCAL: Contrastive Learning for Multimodal Time-Series Sensing Signals in Factorized Orthogonal Latent Space. NeurIPS 2023.
>
> [10] Factorized Contrastive Learning: Going Beyond Multi-view Redundancy. NeurIPS 2023.
>
> [11] The Platonic Representation Hypothesis. ICML 2024.

---

> > ### Comment · Reviewer_DHvL · 2024-11-25
> >
> > I've increased my score according to the authors' feedbacks.

---

> ### Author Response · Authors · 2024-11-25
> **Thanks to Reviewer DHvL**
>
> Thank you for your response and increased score!

---

### Official Review · Reviewer_H7Jq · 2024-11-03

**Soundness:** 4
**Presentation:** 3
**Contribution:** 3
**Rating:** 8
**Confidence:** 2

**Summary:**

In this paper, the authors proposed a novel self-supervised representation learning method for multimodal representation learning. The proposed method disentangles the representation of multimodal information into shared information and modality-specific information, and uses a separate neural network to learn each representation. The authors supported their proposed method through extensive theoretical analysis and proofs. The proposed method was evaluated on both synthetic datasets and real-world multimodal datasets, where the proposed method achieved top performance over baselines in most tasks.

**Strengths:**

1. The paper introduces a new perspective in how to perform effective multimodal representation learning. Specifically, the paper demonstrated that disentanglement of modality-specific and shared information into 2 separate representations can be effective in improving downstream task performance, which seems to be original and novel.

2. The authors provided extensive theoretical justifications and proofs for their proposed approach.

3. The experiments are quite comprehensive. They include both synthetic and real-world datasets, and the proposed method is evaluated against several multimodal SSL baselines on different tasks from drastically different domains. The proposed method achieved top performance in almost all tasks.

4. All experiment details are presented in the Appendix (high reproducibility).

**Weaknesses:**

While the extensive use of information-theory notations allowed rigorous proof of the proposed approach, it isn't really easy for readers who isn't interested in the proofs and just want to learn about the concrete algorithm/methodology. Perhaps the authors should consider including a Pseudocode/Alg block either in the main text or in Appendix to clearly demonstrate the training process.

**Questions:**

Although in most cases DisentangledSSL-both achieves highest performance, there are also several tasks where either  DisentangledSSL-shared or  DisentangledSSL-specific alone achieves top performance (and sometimes by quite a large margin over  DisentangledSSL-both, such as Mustard). Do you observe any trend / suggest any guidelines for when it is best to use each configuration?

---

> ### Author Response · Authors · 2024-11-19
> **Response to Reviewer H7Jq**
>
> First of all, we would like to thank the reviewer for their diligent and insightful review and their very encouraging words. The changes in the manuscript can be found highlighted in magenta. We share our thoughts on the questions asked below.
>
>
> > The author should consider Including a Pseudocode/Alg block to clearly demonstrate the training process.
>
> We thank the reviewer for the suggestion. That indeed is a great point! We have revised the manuscript with an additional pseudocode and a separate table for notations in Appendix I and Appendix J to improve the clarity of our work.
>
>
> > DisentangledSSL-shared or DisentangledSSL-specific alone sometimes achieves top performance.
>
>
> **Please refer to the detailed response about key observations from this table in the consolidated review to all reviewers above.**

---

> > ### Comment · Reviewer_H7Jq · 2024-11-25
> >
> > Thank you for your response! My review remains positive.

---

> ### Author Response · Authors · 2024-11-25
> **Thanks to Reviewer H7Jq**
>
> Thank you for your response and insightful review!

---

### Official Review · Reviewer_nJHk · 2024-11-04

**Soundness:** 2
**Presentation:** 2
**Contribution:** 2
**Rating:** 5
**Confidence:** 3

**Summary:**

The paper introduces a self-supervised learning approach to disentangle shared and modality-specific information in multimodal data. The authors also explain that the optimal case in prior work is not doable based on Minimum Necessary Information (MNI), as the comprehensive analysis of the optimality. The experiments on vision-language and molecule-phenotype retrieval tasks demonstrate its effectiveness.

**Strengths:**

1. The paper presents a rigorous, information-theoretic framework for disentangling data for different domains and modalities under conditions where MNI is unattainable.

2. Based on the theoretical analysis, the authors designed a two-step training algorithm, and provided an optimality guarantee for this proposed method theoretically.

**Weaknesses:**

Weakness:
For the experiments on synthetic data, they only contain the data for unattainable MNI. The synthetic experiments for attainable MNI are also essential here to demonstrate the reliability of the theoretical bound. My suggestion is to add Gaussian noise produced by different variances as the degrees of MNI attainable or unattainable.


1. In Figure 4 (b), the superiority of DISENTANGLEDSSL is not significant compared to other baselines, such as JointOPT. The hyperparameter front of JointOPT is closer to the front of DISENTANGLEDSSL.

2. Table 1 shows the three versions of DISENTANGLEDSSL are unstable. For example, DISENTANGLEDSSL (shared) can achieve promising performance on MOSI while failing on MUSTARD. It would be better to explain why such a phenomenon exists, probably focusing on the main difference between these three versions and properties of multi-modal data.

3. The performance gain shown in Table 2 is quite limited. Could the author explain if small improvements might be important in this domain? In addition, it would be better if the authors could provide the reason for this tiny improvement.

4. The paper does not satisfy the page limitation.

**Questions:**

Please check the weakness section.

---

> ### Author Response · Authors · 2024-11-19
> **Response to Reviewer nJHK**
>
> We thank the reviewer for their time and feedback that helped us improve the work.  We believe that there are a few confusions about our experimental results that have resulted in the given rating. We have endeavored to address your concerns as concretely as possible and ask for your careful consideration of our clarifications. All of the discussions below are added to the revised manuscript in magenta.
>
> We want to emphasize that our work is among the first to tackle the modality gap in multimodal representation learning from the perspective of the unattainability of Minimum Necessary Information (MNI). This, as we believe, is an extremely important contribution, as other methods often assume the attainability of MNI, a condition that almost never holds true in practice.
>
>
> > Synthetic experiments for varying degrees of MNI attainable or unattainable.
>
> This is indeed an important question. Following the reviewer's suggestion, we evaluated our method on data with varying noise levels. As shown in Figure 11 of the revised manuscript, as the noise level decreases, the learned shared representation $\hat{Z}_c$ contains less modality-specific information under the same $\beta$, and the test accuracy on $Y_1$ and $Y_2$ decrease to close to 0.5 even with small $\beta$. Meanwhile, the frontier of test accuracy on $Y_c$ versus $Y_1$ shrinks towards the top left corner, indicating a decreasing expressivity-redundancy tradeoff and easier disentanglement. These trends align well with our theoretical analysis.
>
> Additionally, we refer the reviewer to Figure 10 in Appendix H.1, which experiments on synthetic data with different levels of multimodal entanglement and illustrates the trade-off between expressivity and redundancy in the learned shared representations. In this context, some dimensions correspond to scenarios where MNI is attainable, while others represent cases where it is unattainable. Using such data, DisentangledSSL shows a clear pattern where the learned information plateaus at certain $\beta$ values, which aligns with the property of partially attainable MNI in this synthetic data.
>
>
> > Superiority of DisentangledSSL is not significant compared to baselines in Figure 4(b).
>
> The ideal frontier for the specific information in Figure 4(b) is the one where the specific representation $\hat{Z}_s^1$ achieves low accuracy on the shared target $Y_c$ (y-axis) and high accuracy on the specific target $Y_s^1$ (x-axis). This optimal frontier corresponds to the bottom-right corner of the plot. Our method, DisentangledSSL, produces a frontier closest to this ideal, outperforming all baselines—including DMVAE, FOCAL, and JointOpt. Additionally, Figure 8 in the Appendix presents the frontier for the shared representation $\hat{Z}_c$, where DisentangledSSL’s frontier is significantly superior to that of JointOpt (in this case, the top-left corner is preferred), especially in the middle and left region. These two figures jointly demonstrate that DisentangledSSL more effectively disentangles shared and specific information than all existing methods.
>
> While the experiments on synthetic datasets were conducted to provide a compelling proof of concept supporting our theoretical insights and claims, DisentangledSSL also outperforms existing baselines across real-world benchmarks such as MultiBench and molecule-phenotype retrieval tasks.
>
>
> > Table 1 shows the three versions of DisentangledSSL are unstable.
>
> **Please refer to the detailed response about the key observations from this table in the consolidated response to all reviewers above.**

---

> ### Author Response · Authors · 2024-11-19
> **Response to Reviewer nJHK (continued)**
>
> > The performance gain in Table 2 is limited.
>
> DisentangledSSL significantly outperforms other baselines in Table 2 in most cases, especially in the RXRX19a dataset. For example, DisentangledSSL has on average 8.6% gain in top N retrieving accuracy over the best-performed baseline on RXRX19a.
>
> Further, we note that such a retrieval task (i.e., identifying the molecule perturbation that corresponds to a certain phenotype from a large collection of molecules) is a difficult task, especially when we test on held-out unseen molecules, and it suffers from the noise and potential biases in the dataset. For example, the effect of some drugs may be insignificant, and different drugs may share similar phenotypes [3].
>
> Therefore, even relatively small gains on these tasks are significant. This is also reflected in the results of previous papers on similar tasks (molecule-cell image retrieval) [1,2,3]. For example, in [3], the top-1% recall accuracy of unseen molecules over all molecules is 28.5%, as compared to the best-performed baseline of 27.0%.
>
> [1] Cross-Modal Graph Contrastive Learning with Cellular Images. Advanced Science 2024.
>
> [2] Removing Biases from Molecular Representations via Information Maximization. ICLR 2024.
>
> [3] How Molecules Impact Cells: Unlocking Contrastive PhenoMolecular Retrieval. NeurIPS 2024.
>
>
> > The paper does not satisfy the page limitation.
>
> According to the ICLR 2025 Official Author Guide (https://iclr.cc/Conferences/2025/AuthorGuide), the optional reproducibility statement (Section 6) does not count toward the page limit of 10 pages.

---

> ### Author Response · Authors · 2024-11-25
> **Gentle reminder to respond to our rebuttal**
>
> Dear reviewer nJHK,
>
> As the discussion period is drawing to a close, we wanted to kindly request your feedback on our rebuttal. In our response, we have carefully tried to address all your concerns and also included additional experiments to further demonstrate the strengths of our work. We would greatly appreciate it if you could provide your feedback at your earliest convenience.
>
> Thank you for your time.
>
> Best regards,
>
> Authors

---

> > ### Comment · Reviewer_nJHk · 2024-11-25
> >
> > Thank you for your high-quality rebuttal! My primary concern is that the performance improvement appears to be both limited and unstable. Based on the authors' feedback and comments from other reviewers, even a limited performance gain is considered valuable in this area, but I am still worried that a 1%~2% boost () cannot demonstrate the effectiveness of this method in the practical scenario. Regarding the unstable performance, the updated Table 1 and Figure 5 demonstrate that DISENTANGLEDSSL shows a clear advantage under fair comparison. I would like to increase my score to 5.
> >
> > I am open to any further discussion from the authors, AC, and other reviewers.

---

> ### Author Response · Authors · 2024-11-26
> **Response to the Latest Comment by Reviewer nJHK**
>
> We sincerely thank the reviewer for investing their time and, most importantly, for reconsidering our work based on the revised version. We truly appreciate their effort in helping us refine this draft. We are glad our revisions have addressed most of their concerns, especially the stability of our method.
>
> Regarding the scale of performance improvement, the performance gains are, in general, limited due to the challenging nature of these tasks, as also shown in previous papers. Nevertheless, we see significant performance gain (>2%) for DisentangledSSL in multiple real-world tasks, for example, MUSTARD in MultiBench, and RXRX19a in molecule-phenotype retrieval (where the relative gain of retrieval accuracy is about 10%), as well as in the disentanglement measurement in Table 3.
>
> In addition, we would like to emphasize the theoretical contribution of our paper. We address the challenging real-world setting where Minimum Necessary Information (MNI) is not attainable, with detailed and rigorous analysis in Sections 2.2 and 2.3. This case is underexplored in prior works and as we believe, is an extremely important contribution. While acknowledging the reviewer’s focus on key aspects of our work like empirical performance, we request and would appreciate a re-evaluation from both theoretical and empirical aspects.

---

> > ### Comment · Reviewer_nJHk · 2024-11-27
> >
> > Thank you for your reply. The papers [1][2][3] that you provided to me demonstrate that the relatively small gains can still be considered significant improvements, such as Table 2 in [2]. That makes sense to me.
> >
> > In Table 1, unless MUSTARD, the performance gain brought by DISENTANGLEDSSL is around 2%. I wonder if such a performance gain is significant in the MultiBench. Could you please explain it? Besides, providing me with some papers to support your statement would help. Thank you!
> >
> > [1] Cross-Modal Graph Contrastive Learning with Cellular Images. Advanced Science 2024.
> >
> > [2] Removing Biases from Molecular Representations via Information Maximization. ICLR 2024.
> >
> > [3] How Molecules Impact Cells: Unlocking Contrastive PhenoMolecular Retrieval. NeurIPS 2024.

---

> > > ### Author Response · Authors · 2024-11-27
> > >
> > > We thank the reviewer for their time and thoughtful feedback and for acknowledging the significance of our performance gains for high-content drug screens.
> > >
> > > Regarding the ~2% performance improvement in Table 1 (excluding MUSTARD), we emphasize that even modest gains are meaningful in challenging benchmarks like MultiBench. For reference, the papers below report similarly modest improvements:
> > >
> > > [1] What to align in multimodal contrastive learning? (*concurrent work*) (https://arxiv.org/pdf/2409.07402v1#page=7.0)
> > >
> > > [2] Factorized Contrastive Learning: Going Beyond Multi-view Redundancy (https://arxiv.org/pdf/2306.05268#page=9.0)
> > >
> > > *[2] is already included as a baseline in our original manuscript.*
> > >
> > >
> > > Results are presented in Table 1 (top four rows corresponding to the self-supervised setting, as in our work) in [1] and Table 2 (top five rows corresponding to the SSL setting, like ours) in [2].
> > > However, please note that these works report baseline performance (e.g., CLIP) of around 50% for binary classification, which reflects untuned hyperparameters and is lower than that expected for CLIP embeddings. In contrast, our tuned baselines offer a fairer and more meaningful comparison. Furthermore, their training setup, design choices, and inference protocol differ from those employed in our work. Additionally, our reported numbers for Factor-CL align closely with those in its original paper [2], further supporting the fairness of our evaluation.
> > >
> > >
> > > We hope this addresses the reviewer's concerns and clarifies our contributions. We thank the reviewer again for their valuable insights.

---

> > > > ### Author Response · Authors · 2024-12-01
> > > > **Gentle reminder to respond to our rebuttal**
> > > >
> > > > Dear reviewer nJHK,
> > > >
> > > > As the discussion period draws to a close, we kindly request your feedback on our rebuttal. In our latest response, we have explained in detail the significance of our performance gains on MultiBench, which is demonstrated with references.
> > > >
> > > > We understand you may have a busy schedule, but if you have any follow-up questions or additional concerns, please don’t hesitate to reach out. If our response has addressed your concerns, we would greatly appreciate it if you could reconsider your score and share your feedback.
> > > >
> > > > Thank you for your time.
> > > >
> > > > Best regards,
> > > >
> > > > Authors

---

> ### Author Response · Authors · 2024-12-02
> **Gentle reminder to respond to our rebuttal**
>
> Dear reviewer nJHK,
>
> Since the deadline for reviewer response is today, we kindly request your feedback on our latest response about the significance of our performance gains on MultiBench, demonstrated in detail with references. If our response has addressed your concerns, we would greatly appreciate it if you could reconsider your score and share your feedback.
>
> Thank you for your time and we truly appreciate your insights.
>
> Best regards,
>
> Authors

---

> ### Author Response · Authors · 2024-12-03
>
> Dear reviewer nJHK,
>
> Understanding that you may be busy, and with the author response deadline coming, we would like to take this last opportunity to summarize our discussions and the major updates and clarifications we made during the rebuttal to address your concerns.
>
> 1. The major remained concern of the reviewer is the ~2% performance gain in Table 1 (excluding MUSTARD). We have provided detailed explanations and references in our latest rebuttal to address this concern.
> 2. Other concerns the reviewer initially had, including the stability of model performance in Table 1, and the performance gain in Table 2, have been addressed during the rebuttal.
> 3. We thank the reviewer for engaging and for their care in discussing the empirical results. Since much of our contribution pertains to the theory/method, we would like to also emphasize these contributions of our paper which may not have been fully recognized earlier.
>
> We thank the reviewer for their time and effort, and hope these clarifications address the concerns. If you find our rebuttal satisfactory, we hope that you would kindly consider re-evaluating our work.
>
> Best,
>
> Authors

---

### Author Response · Authors · 2024-11-19
**Consolidated response to all reviewers and the AC**

We thank all the reviewers for their constructive feedback that helped us improve our paper. Based on these comments, we have revised our manuscript, with the changes highlighted in magenta. Below we provide a brief summary of the key revisions:

* We revised the introduction and method section to further demonstrate the motivations and problem settings. We included a pseudocode for our proposed algorithm in Appendix I and a table summarizing the definitions of all variables used in our paper in Appendix J.
* We updated the results of MultiBench in Table 1 and Figure 5 to better clarify the results and communicate the strength of our approach.
* We included additional baselines and related works (CoCoNet and SimMMDG) and conducted ablation studies of varied data noise levels on the simulation setting and different $\beta$ and $\lambda$ on MultiBench.

In response to **reviewer nJHK and H7Jq** about questions around Table 1 of the three versions of DisentangledSSL representations, we provide the following clarification and explanation:

> Clarifications around Table 1 showing three versions of DisentangledSSL

We acknowledge that in the initial manuscript, there was some confusion surrounding Table 1, which may not have effectively communicated the strengths of our approach. To address this, we have added clarifications, improved Table 1, and introduced Figure 5 in the revised manuscript.

In the updated Table 1, we highlight the best results among the three representations—shared, modality-specific, and the concatenation of shared and modality-specific (i.e., both)—for all methods. Further, Figure 5 illustrates the performance of each of these three types of representations learned through DisentangledSSL on MultiBench. MultiBench is a collection of real-world datasets where the importance of shared and unique information varies by task. The performance of each representation (shared/specific/both) depends on the relevance of the shared or modality-specific information to the particular task. We provide detailed descriptions of data modalities and task labels of each dataset in Appendix H.2.

Our key observations are:
* MIMIC, MOSEI, and UR-FUNNY: Combining shared and specific representations leads to superior performance.
* MOSI: Shared representations alone achieve the highest performance.
* MUSTARD: Specific representations capture nuances of sarcasm, such as sardonic expressions or ironic tone, which are crucial for sarcasm-level prediction. This explains the high performance of specific representations in Figure 5.

These result trends align with those reported in Table 4 of FactorCL [1]. Further, we ablate the performance of the shared and specific representations for the baselines in Table 4 in Appendix H.2.

[1] Factorized Contrastive Learning: Going Beyond Multi-view Redundancy. NeurIPS 2023.

---

### Meta-Review · Area_Chair_9epN · 2024-12-19

**Metareview:**

The submission received the ratings of four reviewers, which recommended 5, 6, 6 and 8, averaging 6.25. Given the plenty of competitive submissions in ICLR, this stands at a score above the borderline. The reviewers' concerns focus on the unclear motivation, assumption and some insufficient demonstration in performance. After the substantial rebuttal by authors, most of reviewers considered the concerns well addressed, and one reviewer still maintain some concerns about the performance gain ~2%. After carefully checking the reviewer comments and the authors' feedback, the AC considered that although in real-world case, there are some confounding factors that affect the performance, however, for scientific research, we should allow some tolerance in ideal settings for clear understanding and overcoming problem one by one. Therefore, I tend to recommend acceptance towards the current submission, and hope all advice are well incorporated into the final submission.

**Additional Comments On Reviewer Discussion:**

Please carefully take the advice and some thoughts for improvement about the reviewer rating 5, so that there are some future directions to address the reviewer's concern.

---

### Decision · Program_Chairs · 2025-01-22

Accept (Poster)